# TANGO: Analysis and curation of particles in cryo-electron tomography

**Markus Schreiber** [1,2] **& Beata Turoňová** [1] ✉

Cryo-electron tomography (cryo-ET) enables the visualization of cellular structures in near-native environments, but its potential for spatial analysis has been underutilized due to a lack of versatile tools accommodating biological sample diversity. Available solutions often rely on case-specific or hypothesis-driven approaches, while holistic analyses remain challenging. In this work, we introduce TANGO (Twist-Aware Neighborhoods for Geometric Organization), a framework leveraging point cloud descriptors to analyze spatial arrangements of particles, such as macromolecular complexes, in cryo-ET. By encoding relative positions and orientations of particles as twist vectors, TANGO enables rotationally invariant feature extraction, including structured neighborhood occupancy, lattice topology, or angular deviations. Its modular design and user-friendly interface allow for customization of features, facilitating exploratory analyses of spatial patterns in diverse experimental datasets. With its open-source Python implementation, TANGO advances the ability to decode complex cellular architectures and their functional relationships, offering a particle data analysis tool for the cryo-ET community.

Cryo-electron tomography (cryo-ET) provides structural insights into the landscape of cells in their close to native environment. The basic workflow consists of the acquisition of 2-dimensional (2D) images of a frozen specimen at different tilt angles that are subsequently used for the reconstruction of a 3-dimensional (3D) volume called a tomogram. To gain access to high-resolution information, cryo-ET is often used in combination with subtomogram averaging (STA). STA is based on finding a common alignment among instances of the same cellular feature, such as macromolecular complexes or membranes. Averaging of the aligned instances ("particles") reduces the anisotropy of the resolution and improves the signal-to-noise ratio (SNR), resulting in a structure that can presently reach resolution below 5 Å[1].

However, the range of information obtained by STA extends beyond high-resolution structures. It also provides positions and orientations of the particles of interest, allowing for analysis of their spatial relations to each other and to other cellular features. Such spatial information offers insight into cells, which can be understood as being organized into functional modules: groups of molecules or molecular complexes that carry out specific tasks[2]. These modules do not act in isolation; their coordinated behavior gives rise to the overall molecular sociology of the cell. A key challenge in understanding such cooperation lies in identifying recurring patterns of molecular associations that underlie complex cellular processes. In this context, spatial analysis enhances our understanding of the cellular ultrastructure and can result in broader insights into cellular organization than structural analysis alone[3–7].

In practice, such spatial analyses rely on particle lists that record the positions and orientations of subunits or complexes. These lists can be generated in different ways: historically by manual picking or oversampling of underlying surfaces, and more recently by automated methods such as 3D template matching (TM)[8–10] or deep learning approaches[11,12]. Each strategy has its trade-offs: manual picking is laborious but yields relatively clean lists with known affiliations, whereas TM and deep learning can produce more complete lists but at the cost of high false-positive rates and the loss of explicit affiliation to the underlying object. By affiliation, we mean the assignment of each particle to the larger structure it belongs to (e.g., assigning a subunit to its nuclear pore complex or to a particular microtubule), which is

[1]Department of Molecular Sociology, Max Planck Institute of Biophysics, Frankfurt am Main, Germany. [2]IMPRS on Cellular Biophysics, Frankfurt am Main, Germany. ✉e-mail: Beata.Turonova@biophys.mpg.de

crucial for quantitative analysis of these assemblies. Classification can be used to remove false positives, but this is computationally expensive and does not restore lost affiliation information.

Despite recent advances in STA and particle picking, the tools to exploit particle lists for systematic spatial analysis remain limited to specialized scripts and hypothesis-driven solutions[3,6]. A notable exception is presented by work on statistical particle pattern analysis[13], which aims at the quantitative description of particle clustering based solely on their positions. As for specialized tools, Jiang et al. [14] present a package that is based on the clustering of both relative positions and orientations. While this approach is related to the one presented in our study, it is specific to the study of polyribosomes. To our knowledge, there are no tools for the holistic spatial analysis of particles that are not tailored to a specific use case. This work aims to close this gap by providing a framework based on point cloud descriptors (PCDs)[15]. Point clouds are discrete sets of points in space, such as 3D coordinates from scanning methods that describe shapes[16,17]. They can represent, for example, the positions of atoms[18,19], or be sampled from abstract spaces that constitute virus genomes[20]. The idea of PCDs is to characterize each point by the geometric properties of its local neighborhood and to encode this information in a feature vector. Practically, one first defines the local neighborhood, referred to as the support - often a simple sphere or cube of a given size. For each query point, all points within its support are collected and used to compute the feature vector, or descriptor. Features can be simple, such as the number of points within the support. They can also be more elaborate, such as descriptors based on spherical harmonics[21], capturing properties like surface curvature[22], neighbor distribution, or orientations[23] in a compact way.

Such feature encoding enables reliable and efficient tasks such as object recognition, pattern matching, registration, surface segmentation, and similarity comparison[24–26]. A simple example is the signature of histograms of orientations (SHOT)[27], which captures local surface shape and orientation by computing histograms of normal vectors around a point on a spherical support. These histograms form feature vectors that can be used for pattern matching, either by searching for a specific signature or by clustering similar signatures. In structural biology, point cloud descriptors have been used to analyze protein surfaces and molecular docking through point cloud registration[28].

In cryo-ET, the positions of particles can be considered to form a point cloud. However, what sets these particles apart from classical point clouds is the additional information on particle orientation. In general, point clouds do not need to carry orientation information; if provided, the closest equivalent is the surface normal vectors defined by the underlying surface. In contrast, particles in cryo-ET possess full rotational information and their relative orientations can thus be examined among all particles within a support. Moreover, each rotation specifies a canonical reference orientation. By aligning query particles with this reference, one can ensure that the features computed from the respective supports do not change when the entire system is rotated. In this way, the features are rotationally invariant.

In this work, we explore the potential of PCDs for holistic analyses of particles from cryo-ET. We calculate so-called twist vectors, which encode relative positions and relative orientations of pairs of particles[29] in an inherently rotationally invariant manner. They build the core of the proposed framework called TANGO - Twist-Aware Neighborhoods for Geometric Organization. For each particle, twist vectors are computed for all neighbors within a fixed radius around that particle. These vectors form the basis that allows for a straightforward derivation of other features such as neighborhood occupancy, lattice geometry, angular deviations, and more. The choice of features to be calculated can be chosen in a project-dependent manner. As we show in this work, the outcome of the feature vector computation offers multiple possibilities for further data analysis. First, the results can be used directly to describe particles in a quantitative or statistical manner, for example to determine the presence of cyclical symmetry. Vectors can be further clustered to find affiliations of particles to their underlying objects, as demonstrated on microtubules (MTs) and nuclear pore complexes (NPCs). Clustering can also be used to find outliers in data to either exclude them or to examine them further. We show this on immature virus-like particles (VLPs) of the Human Immunodeficiency Virus type 1 (HIV-1) CA-SP1 lattice and also on capsids of mature HIV-1, where outliers represent pentamers within the hexagonal lattices. Finally, we demonstrate how point descriptors can be used to find patterns among particles, on synthetic data as well as on experimental data.

TANGO is not only a framework, but it also comes with an open-source Python-based implementation which is further enhanced by a graphical user interface (GUI). Its modular design reflects the pipeline outlined above, starting with the computation of twist vectors on a spherical support. To ensure flexibility and support exploratory data analysis, we provide catalogs with options for different supports and features, which are efficiently computed from twist information. Beyond what these catalogs offer, TANGO facilitates the customization of descriptors and supports. The accompanying GUI includes means to visualize particle lists, supports, statistics, and clustering results, further enhancing data analysis.

Overall, the versatility of TANGO allows for a more holistic characterization of spatial arrangements of particles, offers a user-friendly interface, and will be a useful addition to data analysis and processing within the cryo-ET community.

## Results
### Workflow

The approach proposed in this study is based on PCDs. Going through each point in the dataset, data points with positions within a defined query point's support are collected and used for the computation of features that describe the neighborhood's geometric properties. The collection of features yields a feature vector for each query point, also known as a point descriptor. In our setup, point clouds are generated from particle positions, which can stem directly from particle picking or from further iterative refinement using STA. However, apart from positions in 3D space, expressed in the extrinsic coordinate system, we also consider every particle's orientation, which refers to the rotation required to achieve a canonical orientation—the reference used in STA or particle picking. The canonical orientation can be expressed via a particle's intrinsic coordinate frame as further explained in Supplementary Fig. 1a–d.

To use the full potential of the available particle information, and to accommodate the diversity of the analysis needed, we introduce two changes into the classical PCD workflow: First, we consider each query particle in its canonical orientation (the reference orientation) and thus rotate the respective support, i.e., the set of neighboring points, accordingly. This guarantees orientational alignment of the individual supports, ensuring rotational invariance of the descriptors that are computed in this manner. The second change is the hierarchical approach. We first compute the basic twist descriptor (see section on the twist descriptor for details) on larger spherical supports. This descriptor provides essential information on neighborhood distributions and can then be used to derive additional features based on the data and question at hand. Features can be arbitrarily combined, thereby yielding descriptors that best reflect the needs for the given data analysis. In this second step, an arbitrary support can be chosen either from the provided support catalog or custom-made as binary mask. Furthermore, twist descriptors can be used to filter out points within the support based on, for example, their angular distance from the query point. The final step of our framework is the analysis of the computed descriptor(s), which is data-dependent. This step can include quantitative or statistical characterizations of the particles' spatial distribution, the clustering of descriptors in order to find

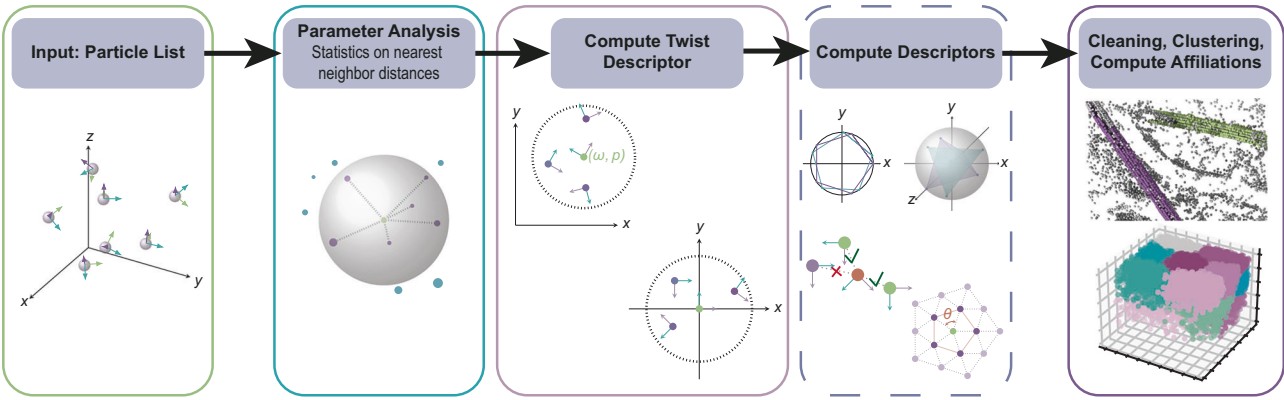

**Fig. 1 | Proposed TANGO workflow.** Starting with a particle list, parameters relevant for the analysis, such as nearest neighbor distances, are extracted. These help to define the support on which the twist descriptor is computed. By default, each support is aligned with the query particle's canonical orientation (middle panel). Using the twist descriptor as basis, additional descriptors might be computed if needed. The final step is the analysis of the descriptor(s), which can be tailored to the task at hand; be it finding patterns in the data using clustering of the descriptor features, identifying outliers, or assigning affiliations to underlying objects.

patterns within the data, or the assignment of particles to their underlying objects. The complete workflow is depicted in Fig. 1.

## Twist descriptor

A twist vector encodes the relative position and relative orientation of a query particle with a given neighbor within its support. This representation captures both directionality and distance (angular distance for orientations, Euclidean distance for positions). Relative positions are represented as vectors in the query particle's own frame of reference, each one pointing towards the neighboring particle's location. Relative orientations are represented as rotation axes scaled by rotation angles. The corresponding rotation encodes how the neighbor's orientation differs once it is expressed within the query particle's frame.

By default, twist vectors are computed within a unified frame of reference: the particle pair is shifted and rotated so that the query particle is aligned with the canonical orientation and placed at the origin of the extrinsic coordinate system. This operation preserves the relative orientation and relative position, ensuring rotational invariance for descriptors based on twist vectors. Such normalization reduces the degrees of freedom when comparing local particle neighborhoods. A full mathematical derivation of twist vectors is provided in the methods section.

The twist descriptor for a query particle comprises the set of twist vectors to all neighbors within its support, along with the distances they encode. Additional practical information, such as the orientational deviation from canonical rotation axes and the in-plane angles of query particles and their nearest neighbors, is also included in this descriptor.

## Particle symmetry

The analysis of relative orientations between symmetric particles can be ambiguous. As an example, a particle showing five-fold rotational symmetry around a given axis has five different choices of rotation matrices representing the same orientation (see Fig. 2a). This range of options needs to be accounted for when considering the orientational differences between two given particles of the same symmetry type. Naturally, one can consider the angular distances between all 25 possible pairs, but the computational expense would grow with more intricate types of symmetry. In our framework, we resolve this ambiguity for two classes of rotationally symmetric particles: cyclically symmetric ones ($C_n$-symmetry for $n > 1$) and those that show symmetry of a platonic solid (tetrahedron, cube, octahedron, dodecahedron, icosahedron).

Our solution is presented in the form of the angular score (see methods section for details), a normalized score that depends on the input symmetry. A notion of dissimilarity between particle orientations is computed as displacement between vertex sets of geometric representatives on which the corresponding rotations act, see Fig. 2a, b. This dissimilarity then yields the angular score for the given symmetry type by normalizing with respect to the maximum dissimilarity. Within the angular score, 0 hints at maximum angular disagreement between two particles of the same symmetry type, whereas score 1 refers to perfect orientational agreement. For example, in the case of five-fold symmetry with a central angle of 72°, a score of 0 would correspond to a displacement of 36°.

## Implementation

The presented framework is implemented in Python and embedded in the cryoCAT toolbox[30] under the module tango.py. This module contains classes for particles, the basic twist descriptor and its derived descriptor classes, features, supports, and filters, as well as catalogs for the latter three. Additional analysis and visualization functionalities tailored to descriptor objects are also provided. A detailed description of the classes is given in the methods section, and an overview of the tool's structure is shown in Supplementary Fig. 2.

The core concept is the twist descriptor, which is initially defined on a spherical support. It is computed from one or more particle lists and a specified support radius. Particle lists from common software packages such as TOM/AV3[31,32], Relion (version 3 and later)[33], STOPGAP[10], and Dynamo[34] can be used as input. It is assumed that particle lists do not contain overlapping particles and thus that each entry in the dataset represents a unique particle. This condition can be ensured as a preprocessing step using cryoCAT's distance-based or shape-based cleaning functions[35].

For the subsequent analysis, a range of supports such as spheres, cylinders, ellipsoids, cones, and tori are provided. In addition, custom binary masks of arbitrary shape can be used. Supports can be refined further by filters that remove particles outside of user-defined property ranges, for example, based on angular distance. From these reduced sets, feature class computes quantitative attributes for each query particle. Unlike the basic twist descriptor, which may contain different numbers of entries for each query point, features always return vectors of equal length, enabling their combination into new, custom descriptors. Custom descriptors can be built from twist descriptors, a set of desired features, and optionally from a different support and filters. Custom descriptors can be easily extended by adding new features (or reduced by removing some of them) and can

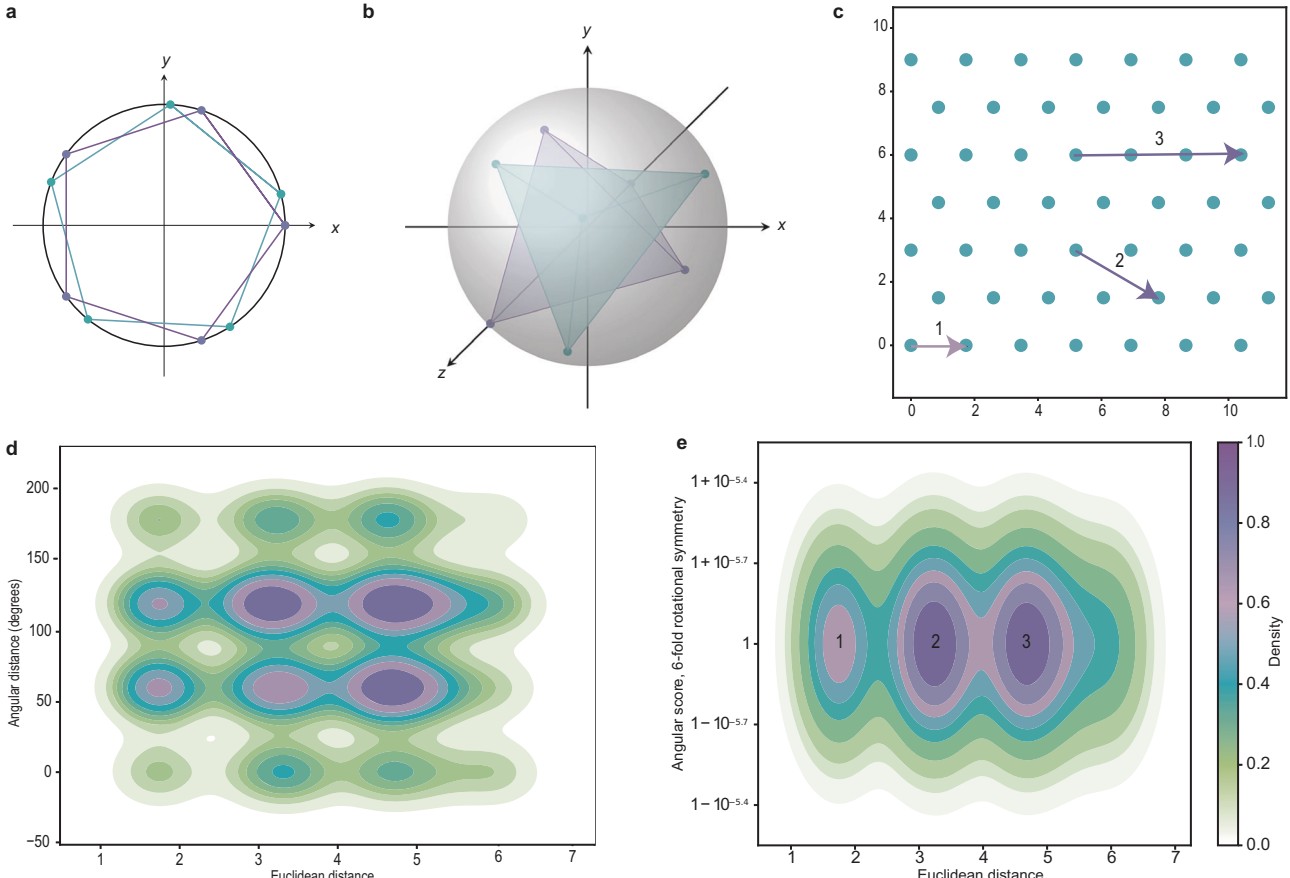

**Fig. 2 | Description of the angular score. a, b** Depiction of geometric representatives of $C_5$-symmetry (**a**) and tetrahedral symmetry (**b**). **c** An example for vertices in a regular, hexagonal lattice in the plane to demonstrate the unambiguous character of the angular score. Each vertex stands for the position of a $C_6$-symmetric particle; orientations are sampled from $C_6$. **d** Kernel density estimation for angular distances measured around concentric shells of each particle from (**c**). Peaks occur at multiples of 60°. White corresponds to the value 0, purple

corresponds to 1. Source data are provided as a Source Data file. **e** By using the angular score instead of the angular distance, density peaks are concentrated around 1 (ideal hexagonal symmetry) at distance steps as indicated by the radii of concentric shells. A particle's first shell (1) is the smallest shell, followed in size by the second shell (2), and the third one (3). The labels 1, 2, 3 correspond to those in (**c**). The depicted data does not account for boundary artifacts. Colors are assigned as in (**d**). Source data are provided as a Source Data file.

also be merged with other descriptors computed on different supports. Besides the custom descriptor, additional descriptors are provided: the piecewise linear (PL) descriptor, the $\alpha$-descriptor, and SHOT (see Supplementary Sections 2.1, 2.2, 2.3 for more details).

Since TANGO is Python-based, it can be used within scripts, interactive Python environments, or Jupyter Notebooks. To make it more user-friendly, we implemented a GUI using the Plotly Dash library[36]. Dash enables interactive data visualization in web browsers and greatly facilitates descriptor analyses by allowing users to easily explore different supports, filters, and features.

Finally, given the versatility of cryo-ET data, we implemented TANGO to allow users to easily extend it with their own features, supports, and filters. TANGO's modular structure ensures that such additions are automatically included in the corresponding catalogs and made available in the GUI.

The full application programming interface (API) documentation, tutorials, and an instructional video for the GUI are provided as part of the cryoCAT documentation.

## Use cases

To demonstrate TANGO's capabilities, several use cases are presented. The parameters defining the supports were chosen based on the physical properties of the underlying data, which were obtained from nearest-neighbor analyses. A complete list of the applied parameters is provided in Supplementary Table 1.

**Object affiliation and cleaning.** Particle lists often encode the positions of subunits within larger complexes such as nuclear pore complexes (NPCs), $\alpha\beta$-tubulin dimers forming microtubules (MTs), or subunits forming viral lattices (e.g. the immature HIV-1 CA-SP1 lattice). Strategies for generating such lists manually, while laborious, typically yield relatively clean data with clear affiliations to their parent structures, enabling straightforward downstream analyses such as measuring NPC radii[37]. By contrast, modern approaches such as 3D template matching (TM)[8–10] and deep learning[11,12] involve a trade-off: using permissive thresholds leads to more complete particle sets but with higher false-positive rates, while more strict thresholds reduce false positives but result in incomplete datasets. In both cases, explicit affiliation information is lacking. Here, we demonstrate how TANGO can address these issues by recovering affiliations directly from positional and orientational information while simultaneously cleaning particle lists. In the three use cases discussed below - NPCs, MTs, and VLPs - all necessary information was contained in the twist descriptor, so no additional computation was required.

Affiliation information can be derived directly from twist descriptors, since they encode nearest-neighbor relationships that can be interpreted as a graph. In this graph, particles are nodes and two particles are connected via an edge if there is a twist vector associated with the pair. A twist descriptor thus decomposes into connected components, where nodes from one component are affiliated. We refer to this topological way of affiliating particles as proximity clustering.

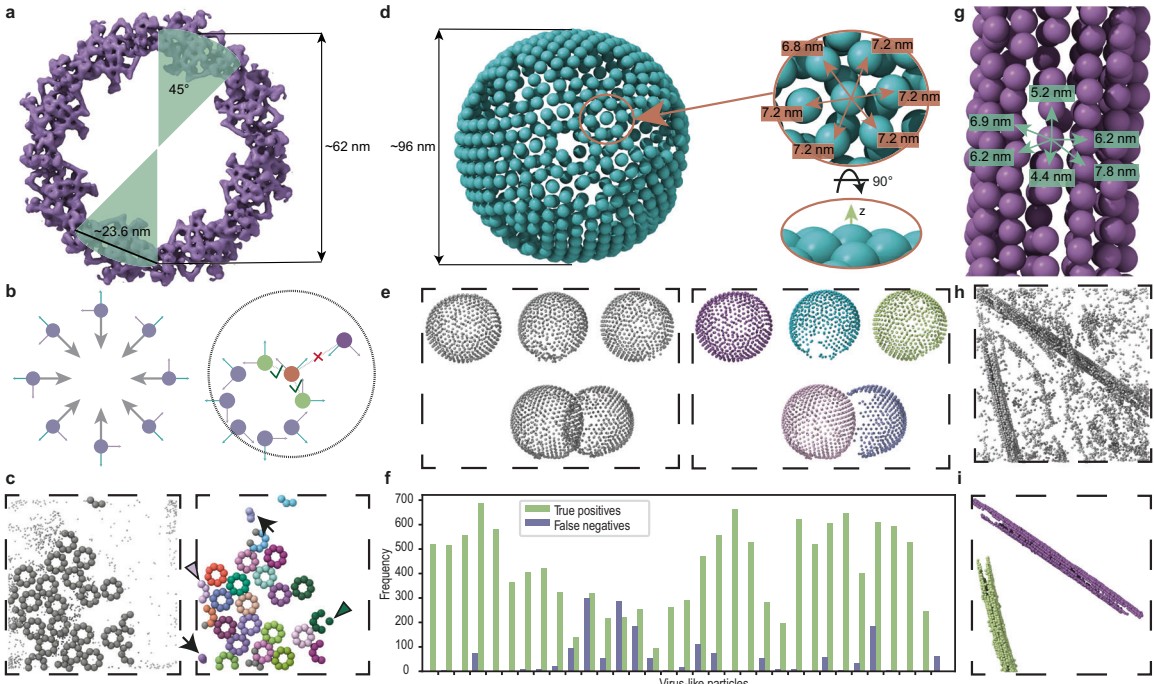

**Fig. 3 | Cleaning of particle data and computation of affiliations. a** Cytoplasmic ring from[6] visualized using EMD-51626. An angle of 45° associated to a given sub-unit (SU) is shown to indicate the complex's $C_8$-symmetry. **b** The ideal positions of the asymmetric units of the nuclear pore complex (NPC), showing eight-fold symmetry, together with their orientation (left). The orientation is used to shift particles along their intrinsic -$x$-axes by a value close to the NPC radius (right). This allows to reduce the initial spherical support for the twist descriptor that is further cropped to a cylindrical support (dashed line) followed by filtering out particles that deviate from the expected relative orientation. **c** Particles from the uncleaned particle list are placed back into a tomogram. Particles from the manually curated ground truth are represented with larger spheres (on the left). Small spheres represent positions that were originally labeled as false positives. Results of cleaning and affiliation computation for NPCs are depicted on the right. Different colors are associated with different NPCs. Arrowheads point to the particles that were not in the ground truth but were correctly recovered by our method. Arrows point to wrongly assigned clusters. **d** Representative CA-SP1 lattice of immature HIV-1 virus-like particles (VLPs) from the EMPIAR-10277 dataset, shown with subunit distances and a side view indicating the intrinsic $z$-axis (corresponding to the sur-face normal). **e** Particles representing CA-SP1 lattices of immature HIV-1 VLPs placed back into a tomogram (left). The assigned color-coded affiliations of subunits to their respective VLPs are shown on the right. **f** The results of the affiliations for the complete HIV-1 dataset are shown for each VLP separately. Source data are pro-vided as a Source Data file. **g** Microtubule (MT) data from[9] with indicated subunit distances. **h** Particles from a particle list obtained from template matching (TM) using a low-constrained cross-correlation (CCC) threshold to ensure completeness of the data. **i** The result from clustering based on spatial proximity shows two distinct clusters corresponding to two MTs present in the data. The different colors correspond to the affiliation of each particle to its corresponding MT.

Treating twist descriptors as graphs also provides a strategy for cleaning: pairs of particles (edges in the graph) can be removed based on the expected underlying geometry. In the cases considered here, this means requiring the twist descriptor to reflect that NPCs are embedded in the nuclear envelope, as well as cylinders (MTs) and spheres (VLPs) being locally flat, meaning around each point, they can be approximated by a (tangent) plane within a small radius around that point. Cylindrical supports were chosen in all three cases to capture pairs of particles that were close to the same plane. The cylinder's axis was aligned with a normal vector to the respective surface. Further pairs were excluded from the twist descriptor by restricting the allowed angular distance (reflecting the eightfold rotational symmetry of NPCs), or by requiring that the axis of their relative rotation remained close to the surface normal.

To assess the performance of cleaning and affiliation, we used the $F_1$-score[38], a metric that combines the rate of correctly retained parti-cles with the penalties for false positives and false negatives, providing a single measure of how reliably the cleaned and affiliated particle sets reflect the ground truth (see methods section for more details).

For the NPC structure, we worked with a dataset from ref. 6 (Fig. 3a). The input particle list was generated from GAPSTOP™ score maps from the original study, produced with a template containing a subunit of the NPC's cytoplasmic ring (CR). In total, 93,665 asymmetric subunit (SU) candidates were obtained from these score maps, which then needed to be cleaned and affiliated to their respective NPCs. A manually curated particle list containing 1960 SUs from 322 NPCs served as ground truth for the subsequent analysis. The list included the affiliations of each SU to its respective NPC.

As a preprocessing step, the particle list was cleaned from spatial duplicates by applying shape-mask cleaning. This reduced the number of particles to 63,187. Next, the particles were shifted in the intrinsic -$x$-direction by a distance close to the NPC radius (Fig. 3b). This shift ensured that SUs belonging to the same NPC clustered more closely, which allowed for smaller spherical supports, reduced dense regions of false positives, and improved computational efficiency (see Sup-plementary Fig. 3).

The cleaned and affiliated set was then obtained using proximity clustering, which performed both tasks simultaneously by requiring at least three subunits to form an NPC. This reduced the number of particles from 63,187 candidates to 2341 affiliated SUs. The affiliation achieved a median $F_1$-score of 0.967 across all tomograms, high-lighting the high precision of positive assignment. An example of the cleaning is shown in Fig. 3c. Small examples of false positives and false negatives can be observed. By visual inspection, two particles in this example were correctly affiliated despite not being part of the ground truth.

In the case of VLPs of immature HIV-1, we focused on the corre-sponding CA-SP1 SUs from the EMPIAR-10277 dataset (see Fig. 3d).

Unlike in the case of NPCs, the particle list we used was already clean in the sense that it did not contain any false positives, yet the particles were lacking affiliation information. The latter was determined using the proximity clustering. The affiliation reached a median $F_1$-score of 0.97 taken over all 32 VLPs in the dataset (see Fig. 3e, f). The lowest $F_1$-score of 0.48 was achieved for a VLP for which 139 true positives were found, versus 300 false negatives. This discrepancy could be improved by choosing larger support radii to account for the data sparsity of individual VLPs. For this use case, however, we decided on homogeneous, robust parameters that would benefit most VLPs.

A similar approach of computing affiliations was applied to MTs (see Fig. 3g). A tomogram with two MTs (from[9]) was used to generate a score map, following the original study that employed a short MT segment as the template. We extracted a particle list of $\alpha\beta$-tubulin dimers using an absolute CCC-score threshold of 0.015 to ensure completeness of the data. The extracted positions can be seen in Fig. 3h.

In total, the particle list contained 6997 particles. After computing and filtering the twist descriptor, proximity clustering yielded two groups of particles with 1174 and 974 members, respectively. Due to the lack of ground truth, the results were first evaluated based on visual inspection (see Fig. 3i) which showed that the affiliations were correct and that all obvious false positives had been removed.

To further validate the procedure, we also performed STA on three datasets: the combined particle set, the cleaned list, and the list of removed particles. Initial averages were generated from TM positions and refined through 10 iterations of alignment (see Supplementary Fig. 4 and Supplementary Table 2). As expected, all initial averages resembled the template to some degree. For the combined and removed lists, however, the maps degraded with further alignment, while the cleaned list improved, supporting the effectiveness of the cleaning. Some resemblance to microtubules remained in both the combined and removed sets; in the former case this is expected due to the presence of true particles in the list, while in the latter case it may indicate that a small fraction of true particles were marked as false negatives.

**Lattices.** Pleomorphic assemblies such as protein filaments, coated vesicles, or some viral capsids are omnipresent in biological samples. They are built as non-covalent compounds from numerous independently synthesized molecules that serve as SUs of the final lattice, which covers the underlying geometry[39]. Analyzing regular, repeating patterns in such a structure is relatively straightforward. In contrast, spotting irregularities or deviations from this order can be more challenging, yet these are often the structures that are worth exploring in more detail.

**Mature HIV-1 capsids.** Here, we demonstrate TANGO's ability to identify pentamers on the hexagonal lattice forming capsids of mature HIV-1[6]. The mature HIV-1 capsid has a conical shape containing in total 12 pentamers located at regions of high curvature, hidden among hexameric SUs that form the rest of the lattice[40], see Fig. 4a.

The input particle list contained positions of both hexamers (1510 in total) and pentamers (57 in total) for 11 capsids. Note that all capsids were incomplete, i.e., were missing a certain number of both hexameric and pentameric SUs. The incompleteness likely resulted from technical limitations, although a biological contribution cannot be fully excluded.

At first, all eleven capsids were considered, for which the initial support was truncated to focus on the first shell only. A visualization of the notion of shells and of central angles is shown in Fig. 4b. For the truncated support, the $\alpha$-complex descriptor was computed. This provided various statistics such as the number of edges, the median and STD of the central angles (where the expected value is 60° for regular flat hexagons), Euler characteristics, and the link type (see

Supplementary Section 2 for an explanation of these additional features). Among these options, a subset of significant features was determined using principal component analysis (PCA). Applying PCA to all computed features revealed that to explain more than 95% of variance, it suffices to consider the median of central angles, their STD, and the number of vertices. Using those three features we identified pentamers in 70.8% of cases. Figure 4c shows the results per capsid.

Secondly, a single lattice's regularity was investigated using the angular score $\sigma_V$ (see Eq. 12). Over a given shell, the angular score was computed between the central particle and every particle in that current shell. When the rotational symmetry of lattice particles is unknown, the angular score can be computed under the assumption of $C_n$-symmetry for a range of $n$ values. In the present case (Fig. 4d; Supplementary Table 3), the lattice shows the strongest signal for $C_6$-symmetry, so the corresponding angular score was added to the twist descriptor and used for subsequent analysis. Once the symmetry is established, lattice regularity can be assessed by tracking the mean angular score over shells of increasing radii. As shown in Fig. 4e, the hexameric lattice exhibits the highest degree of regularity in the first shell around each particle, which gradually decreases with distance, likely reflecting the capsid's curvature.

**Immature HIV virus-like particles.** A similar analysis was performed for the VLPs of immature HIV (same data as in the section on object affiliation and cleaning). The twist descriptor was computed, including the angular score, on spherical supports. This was followed by the analysis of individual shells. The mean angular score of the first shell around each particle shows that about two-thirds of the particles have highly regular hexagonal shells (mean angular score > 0.8; Fig. 4f). The information on angular similarity can be applied to subdivide the data into regions of different levels of angular agreement (see Fig. 4g), thus highlighting irregularities for further investigation. To test whether a low angular score indeed reflects hexamers with disrupted neighborhoods or misaligned orientations, we performed STA on five classes corresponding to different degrees of angular similarity. For a fair comparison, the number of particles in each class was reduced to the smallest occupancy, ensuring equal particle counts across classes (the full parameter setup is listed in Supplementary Table 5). Already in the initial averages, the classes with lower angular scores showed visibly disrupted densities compared to those with higher scores (see Supplementary Fig. 5a). To further challenge the robustness of the classes, we randomized the in-plane angles and ran the alignment. While the highest-scoring class successfully recovered the hexamer and its surrounding hexameric lattice -even without enforcing $C_6$-symmetry - the lower-scoring classes failed to reconstitute the structure (Supplementary Fig. 5b, c).

These results demonstrate that the angular score is a reliable indicator of lattice regularity, with lower scores reflecting structural irregularities. Such irregularities may arise from broken regions of the lattice, as shown here, or from local changes in geometry, such as alternative symmetries within an otherwise regular arrangement as shown on the mature HIV-1 capsids. This makes the score broadly applicable to detecting both defects and structural variants.

**Pattern recognition.** Molecular complexes in cells often interact with each other and may form recurring spatial patterns that help us better understand these interactions. Such patterns can be revealed using point descriptors: either in an exploratory way, where dimensionality reduction and clustering methods highlight trends in the data, or in a targeted way, where supports and features are tailored to specific hypotheses about expected structures.

**Pattern recognition in synthetic chromatin data.** To assess the ability of PCDs to uncover patterns within data, we created synthetic data simulating an arrangement of nucleosomes within chromatin.

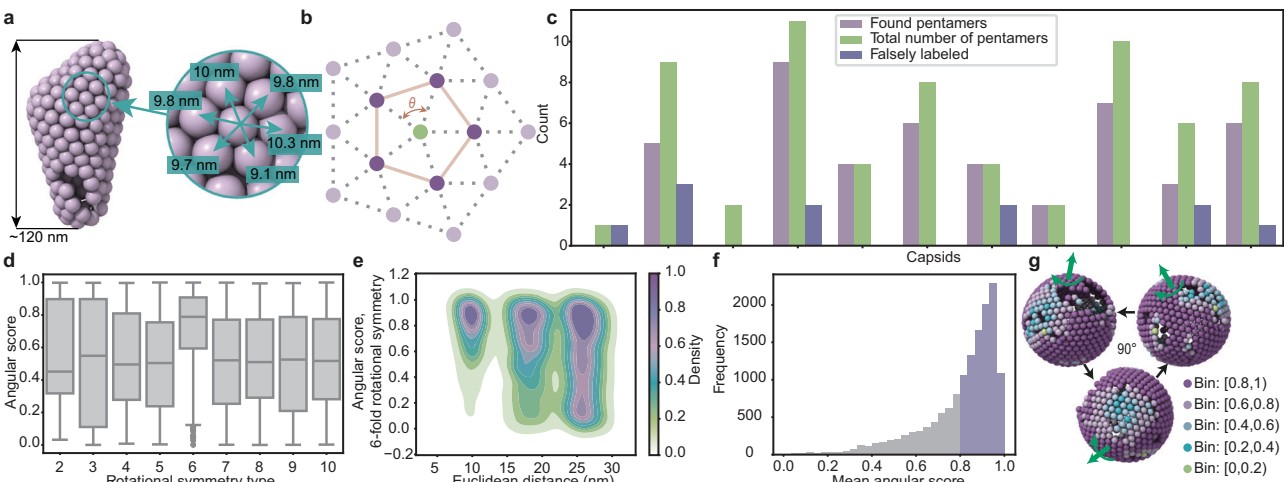

**Fig. 4 | Analysis of lattices. a** A representative capsid of the mature HIV-1 dataset with indicated physical distances between its subunits. Note the incompleteness of the capsid. **b** A two-dimensional depiction of a lattice structure. Given a query vertex (green) in the lattice, the edges connecting its incident vertices form the link of that vertex (orange edges). Alongside statistics describing the central angles $\theta$, the link's topology and architecture (see equations 14 and 15 in the supplementary information) yield features to cluster with respect to the immediate shell of each particle. **c** Results of pentamer labeling based on link topology and geometry. The total number of pentamers refers to the annotated ground truth. Within each capsid, some particles were correctly identified as pentamers (found pentamers), while others were incorrectly labeled as pentamers despite being hexamers in the ground truth (falsely labeled). Source data are provided as a Source Data file. **d** Symmetry analysis of the lattice shows a trend suggesting $C_6$-symmetry. Note that this result was achieved without imposing any preferred symmetry during the analysis. For the comparison of the data collected under the assumption of $C_6$-symmetry with data collected under the assumption of $C_n$-symmetry, the two-sided Mann-Whitney $U$ test was applied to the angular score collected in the respective cases. The angular score was computed for a total of 844 particles in the given lattice. From left to right, $C_2$ through $C_{10}$, the minima of the depicted box plots are 0.03, 0.00, 0.01, 0.00, 0.00, 0.00, 0.00, 0.00, 0.00. In the same order,

the first quartiles are 0.32, 0.11, 0.28, 0.24, 0.59, 0.25, 0.29, 0.21, 0.28. The medians are 0.45, 0.55, 0.50, 0.50, 0.79, 0.52, 0.51, 0.53, 0.52. The third quartiles are 0.90, 0.90, 0.81, 0.75, 0.91, 0.77, 0.77, 0.79, 0.78. The maxima are 1.00, 1.00, 1.00, 1.00, 1.00, 1.00, 0.99, 1.00, 1.00. The $p$-values of the comparison between the $C_6$ assumption and the $C_n$ assumption ($n = 2, 3, 4, 5, 7, 8, 9, 10$) are 8.4e-19, 4.0e-20, 3.6e-35, 8.6e-51, 1.5e-49, 4.5e-46, 7.6e-47, 9.5e-46. The $U$ test values are 2.7e5, 2.6e5, 2.3e5, 2.0e5, 5.1e5, 5.0e5, 5.0e5, 5.0e5. These results are detailed in Supplementary Tables 3, 4. Source data are provided as a Source Data file. **e** Analysis of the lattice regularity for $C_6$-symmetry based on angular similarity between the particles surrounding each query particle (i.e., all hexamers) in concentric lattice shells. The kernel density shows a trend towards high angular similarity ($\sigma_V \approx 1$) between $C_6$-symmetric particles, where regularity decreases over shells of larger radii, likely due to the capsid's curvature. White corresponds to the value 0, purple corresponds to 1. Source data are provided as a Source Data file. **f** The distribution of the mean angular score over the immediate shells for the hexamers of CA-SP1 forming lattices of immature HIV-1 virus-like particles (VLPs). It shows a trend towards high angular similarity. Source data are provided as a Source Data file. **g** A representative VLP of immature HIV-1 with its hexamers colored based on varying degrees of hexagonal regularity. Axes of rotation are indicated.

Nucleosomes can be arranged in different structures and forms, for example, di- and trinucleosomes as described in ref. 41. Such structures, together with the mononucleosome structure (EMD-63079), the canonical orientation of which is indicated in Fig. 5a, served as input for the generation of the synthetic particle data (see methods section on synthetic chromatin data for details). The resulting data consisted of six helical arrangements (with 12 particles each), 350 stacked arrangements, and 340 trinucleosomes. A total of 3200 mononucleosomes were added in varying number densities, see Fig. 5b.

This synthetic list was used as the unperturbed baseline for further analyses. To assess robustness, we further corrupted the data with varying levels of positional and orientational noise (see methods section on synthetic chromatin data). Five levels of positional noise were added, resulting in deviations up to ~0.8 nm for level 1 and up to 4 nm for level 5. Orientational noise levels were sampled up to 1, corresponding to angular deviations up to 60 degrees as indicated in Fig. 5c.

Analysis of angular distances in the initial support pointed to short angular distances which likely corresponds to stacked nucleosomes (Fig. 5d). Using cylindrical supports aligned with the intrinsic $y$-axis in the positive direction (Fig. 5a), the adapted twist descriptor detected 100% of stacked particles. It also classified 66.7% of trinucleosome particles as stacked—consistent with the fact that two out of three are arranged this way (Supplementary Fig. 6c)-- and 83.3% of particles belonging to the helical arrangements, reflecting that only 10 of 12 nucleosomes per the helical arrangement were captured with this approach (Supplementary Fig. 6a). Inspection showed that the two missing nucleosomes were consistently the same stacked pair with

slightly larger angular distance. Only 1% of mononucleosomes were classified as stacked ones, some of which could have been part of stacked configurations.

Overall, the percentages concerning the structures of interest (stacks, trinucleosomes, helices) decreased when corrupting the list with increasing noise, while the percentage of classified mononucleosomes remained low (Fig. 5e). The highest fluctuation in quality can be observed for particles in the helical arrangements, which might be attributed to their low number. The other pattern of interest was dinucleosomes—pairs of nucleosomes connected via linker DNA, as seen in trinucle-osomes (Fig. 5f). To capture such connections, we used conical supports aligned with the $z$-axis, which approximately corresponds to the entry and exit path of DNA. We then probed for common axes of rotation among neighboring nucleosomes, revealing dominant axes (Fig. 5g). Based on this, we defined a custom descriptor combining three features: Euclidean distance, angular distance, and deviation from the most dominant common axis. These features were clustered using k-means (Fig. 5h).

In the unperturbed data, the k-means algorithm revealed a cluster containing 100% of helix-labeled particles and all trinucleosome-labeled particles. This cluster also included 22.7% of mononucleosomes and 21.7% of stacked nucleosomes. Visual inspection of the latter two groups showed arrangements with orientations resembling dinucleosomes. However, since nucleosomes were randomly added during data generation, the exact number of dinucleosome arrangements is unknown, making it difficult to quantify the accuracy of these findings. In case of experimental data, one way forward would be to

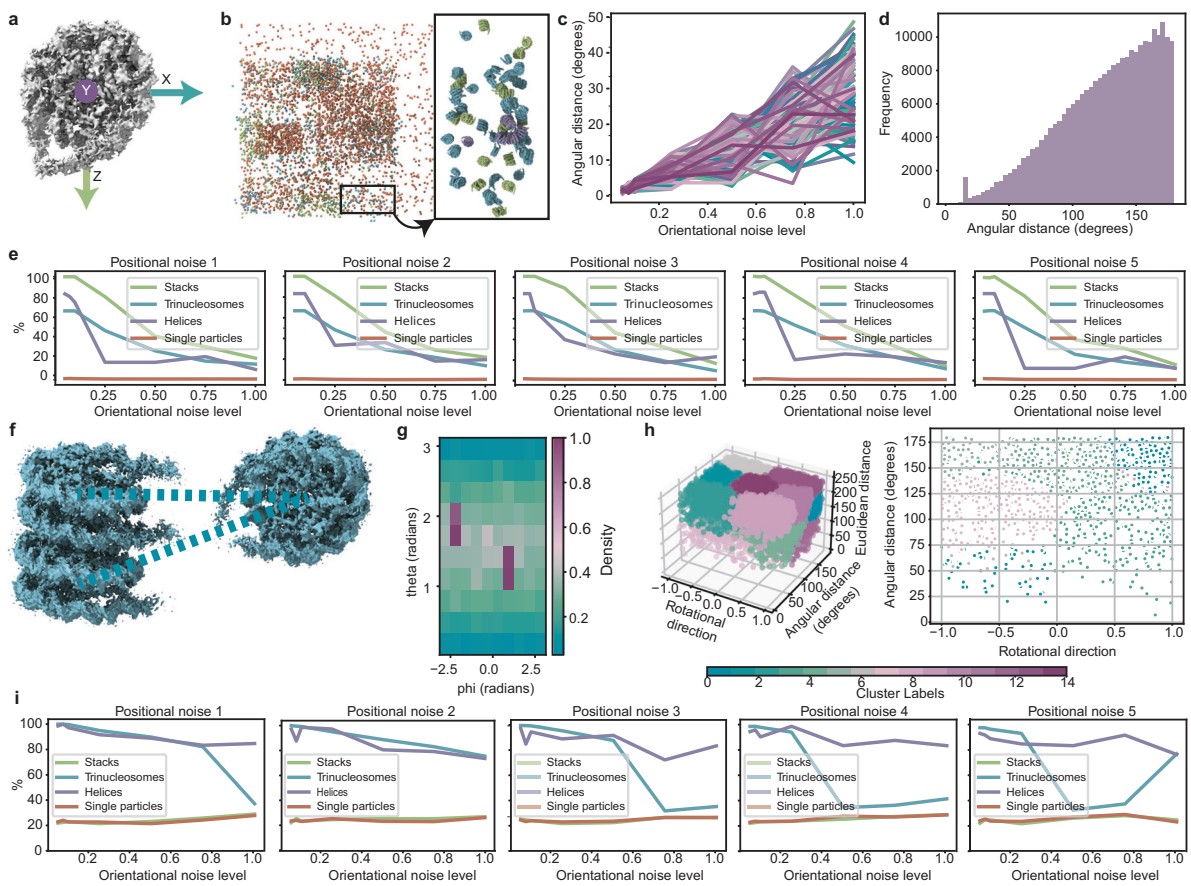

**Fig. 5 | Spatial analysis of synthetic chromatin data. a** A nucleosome (EMD-63079) shown in its canonical orientation with indicated intrinsic coordinate frame. **b** Overview of the synthetically generated particle list with varying local densities and inserted structures of stacked nucleosomes (EMD-13365, light green), trinu-cleosomes (EMD-13363, blue), helically arranged tetranucleosomes (EMD-2601, lavender), and mononucleosomes (EMD-63079, orange) (left) and a rotated close-up of the indicated region (right). Note that in the close-up, we removed the mononucleosomes to improve visualization. **c** The effect of different levels of orientational noise on angular distance. The graph presents a sample of 100 ran-domly generated rotation matrices. Source data are provided as a Source Data file. **d** Histogram of angular distances between query particles and their neighbors within the initial spherical support showing a clear peak at the angular distance of 15°. Source data are provided as a Source Data file. **e** The performance of stack-classification with 5 levels of positional noise shown over different levels of orientational noise. Percentages are in reference to the total amount of the

respective structure. Source data are provided as a Source Data file. **f** A nucleosome model (EMD-63079) placed into the trinucleosome structure (EMD-13363). The dashed lines show dinucleosomes. **g** A polar histogram with scores indicating relative occupation of bins by axes of rotation as computed from the twist vectors computed on a conic support extending in intrinsic *z*-directions. This analysis yielded the feature we called rotational direction. Teal corresponds to the value 0, mauve corresponds to 1. Source data are provided as a Source Data file. **h** Overview of 15 clusters from k-means clustering applied to rotational direction, angular distance, and Euclidean distance (left). The right panel shows the section where Euclidean distance ≈24.5 nm. Source data are provided as a Source Data file. **i**. The performance of dinucleosome classification with 5 levels of positional noise is shown over different levels of orientational noise. Percentages are in reference to the total amount of the respective structure. Source data are provided as a Source Data file.

perform STA on query particles from such detected pairs using orientation-guided masks, which could help validate the results.

When noise was introduced (Fig. 5i), helix classification proved most robust, whereas trinucleosome classification dropped at posi-tional noise levels above 4 and orientational noise levels above 0.75.

**Pattern recognition in ribosome data.** Ribosomes are central to protein synthesis, and recent advances in STA resolution have enabled the study of their conformational states within cells. Beyond transla-tional states, several studies have also examined their spatial organi-zation in the cellular context[3,4,42,43]. Here, we demonstrate how such observations can be addressed using point descriptors. We analyzed data from Xing et al. [4] who studied eukaryotic ribosomes (80S) in human cells under unperturbed conditions and after treatment with the active-site drug Homoharringtonine. Consequently, our input consisted of two particlelists from untreated and treated cells.

Following the initial analysis using twist descriptors, we computed SHOT descriptors for both conditions and focused our analysis on the

three most frequently observed footprints in each condition (see methods section on pattern detection in ribosome data and Supple-mentary Fig. 7 for more details). To characterize these footprints in terms of relative orientations, we first identified their most common axis of rotation using 2D polar histograms. Next, rotations around this axis were quantified by binning angular distances into five ranges. Representative ribosome pairs from each range as well as the 2D polar histograms are shown in Fig. 6.

In all but one footprint, the most common axis of rotation was the same across conditions. The exception was the third footprint, where untreated and treated samples showed distinct preferred axes (Fig. 6c, f).

In comparison with Xing et al. [4] three configurations of pairs of ribosomes within polysomes were described: top-top (t-t), top-down (t-d), and top-up (t-u). Representatives of all three classes could be found using the exploratory approach described here, as well as clas-ses that were not conclusively categorized (see Fig. 6). Interestingly, the mutual orientation among the pairs belonging to the bin

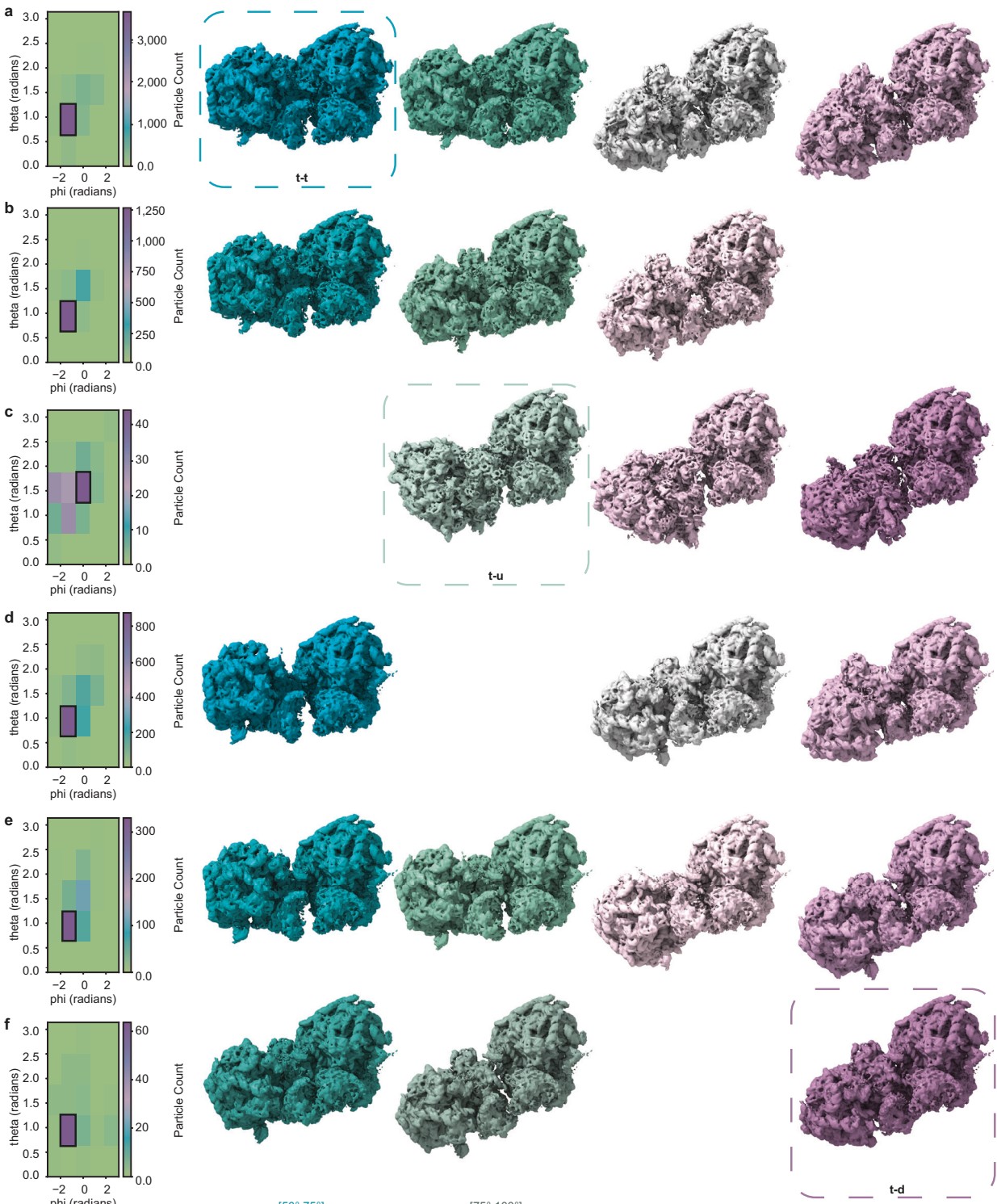

**Fig. 6 | Spatial analysis of ribosome data. a–c** Untreated condition. Each row shows a signature of histograms of orientations (SHOT) footprint, with frequency decreasing from **a** to **c** (**a**: 4,968 pairs; **b**: 1,695; **c**: 140). Left: polar histograms of relative rotation axes; the densest bin (most common axis) is outlined in black. From this bin, pairs were further sub-binned by their rotation angle around that axis (angular distance ranges indicated below). Right: representative ribosome pairs from these ranges, rendered directly with the ribosome map (EMD-16721; no sub-tomogram averaging performed); the query particle is on the right. **d–f** Treated condition, analogous to **a–c** (**d**: 1,554 pairs; **e**: 635; **f**: 91). Representative configurations from[4] are marked t–t (**a**), t–u (**c**), and t–d (**f**). Colors encode angular distance; different hues denote distinct SHOT footprints. Source data are provided as a Source Data file.

highlighted in Fig. 6f, remarkably resembles the stalling disomes described in ref. 42. To validate our findings, STA was performed for all cases as depicted in Fig. 6. More precisely, we used the alignment from the original study to extract and directly average the pairs. Due to a low number of particles, the extraction was done on deconvolved tomograms with a binning factor of two (with pixel size 2.446 Å); no further alignment was performed. A selection of the outputs is shown in Supplementary Fig. 8. In the untreated case, strong densities

surrounding an averaged ribosome pair hint at the found pair being part of polysomes, whereas in the treated condition, the averaged maps hint at secluded pairs. These findings align with the results presented by Xing et al. [4]. STA of the pairs from the most occupied bin of the third footprint of the treated condition (Fig. 6f) was overlayed with the map of the stalling disome (EMD-10398) (Supplementary Fig. 9). While the overall mutual orientation of the pairs aligns well with the disome map, the resolution of the average is not sufficient to conclusively confirm that the pairs form stalling ribosomes. Nevertheless, it shows the potential of our method to find rare occurrences within data.

## Discussion
### Limitations
While powerful, the proposed approach has its limitations. The most notable one in the context of cryo-ET concerns the possible incompleteness of data. Due to many artifacts present in tomograms and imperfections in both particle picking approaches and STA, input particle lists rarely contain all particles that are present in tomograms. While this issue does not affect statistical analysis, finding patterns or reliably assigning affiliations might be negatively affected. For instance, finding longer patterns of nucleosomes, such as the presented helically arranged tetranucleosomes, would fail in the case of missing particles and might skew the overall picture of chromatin arrangement by suggesting the existence of only shorter patterns. One way of dealing with such scenarios is to implement a hierarchical approach in which one would first find the shorter motifs and in the second step tried to connect them into longer ones based on their proximity and orientation.

Another challenge comes from noise caused either by particle misalignment or by inherent imperfections of cellular components – due to the inherent flexibility of biological matter, it rarely occurs that reality corresponds to geometrical expectations. For instance, the SUs of NPCs form a regular octagon and thus the central angle should be 45°. However, even for well-aligned SUs, the central angle deviates from this ideal value, making it difficult to distinguish between physiological and misalignment-related effects in the data. One thus has to be careful when discarding data based on their geometry. It is advised to leave room for some deviations from the expected values. For this reason, TANGO is able to corrupt synthetic data with positional and orientational noise to assess the performance of classification under such conditions.

For point clouds in general application, these issues can usually be mitigated by the use of different descriptors and statistical means (for instance using histograms on the support, instead of direct values). Unfortunately, such approaches are less suited for cryo-ET data due to its sparsity. We propose tackling noise-related issues by appropriately setting filtering parameters and providing a GUI to make this optimization more straightforward. As for data incompleteness, we currently do not offer an implemented solution; however, it should be possible to use this framework to actually identify more particles within the primary data and thereby reduce incompleteness. For instance, to search for trinucleosomes, one can identify footprints where one of the three nucleosomes is missing and search for the corresponding CCC peaks in TM score maps or use those positions and expected orientations as a starting point for STA.

Another possible future direction that could make this framework more robust to noise and incompleteness is the application of neural networks, as they have already proved themselves very useful in the classic point cloud descriptors field[44]. In cryo-ET, the use of deep learning approaches is still very limited in comparison to other areas. The main reason for that is the lack of annotated experimental data that is required for the reliable training of models. While it is difficult to generate data that would truthfully mimic noise properties in the case of image processing tasks, it should not be a problem to generate enough training data in the case of particle lists, as demonstrated in the synthetic nucleosome case.

## Conclusions
The rich nature of particle data from cryo-ET comes from positional and orientational information. Here, we applied a framework based on point cloud descriptors to analyze the geometric interplay between particles. The core of our framework relies on the use of twist descriptors, which provide an elegant and efficient way to describe these mutual relationships and offers a versatile basis for further analysis.

We showed that the twist descriptor alone is sufficient for geometric cleaning of particle lists that contain either misaligned or falsely picked particles (the latter typically caused by automated particle-picking methods). Using our approach, we successfully removed false positives from lists stemming from TM of $\alpha\beta$-tubulin dimers as well as of the asymmetric unit of the NPC's cytoplasmic ring. In the latter case, our approach even recovered particles that had been erroneously removed during manual cleaning. Furthermore, focusing on orientational aspects yields built-in means of separating connected components, which allowed us to affiliate the particles with their underlying objects. Naturally, this approach to automated cleaning and affiliation computation is only suited for particles that assemble in a known geometry, such as large macromolecular complexes with known symmetry, or pleomorphic assemblies.

We further demonstrated the capabilities of features built on the basic twist descriptor to analyze highly structured particle lists – ones whose particles assembled in lattices. Different degrees of local incompleteness of these lattices can be investigated using descriptors of particle neighborhoods, as well as different lattice symmetries. We showed this on incomplete capsids of mature HIV-1, where a descriptor suited for regular lattices was used to identify pentamers within hexagonal arrangements, as well as local incompleteness within neighborhoods. A feature designed to account for orientational ambiguity among symmetric particles was used to identify irregularities within CA-SP1 lattices of immature HIV-1 VLPs.

Finally, we showed the potential of our method for the analysis of soluble particles, such as ribosomes. By applying an exploratory approach to ribosome data that can be partitioned into two experimental conditions, several common characteristics between those cases can be identified and statistically recorded. Such findings can be reinforced using STA to obtain structures from patterns or for fast classification. We thus demonstrated the potential of our framework in the context of geometric characterization, which may facilitate the comparison of experimental data exposed to differing conditions, such as wild types and mutants.

Although diverse in character, all analyses were conducted following the suggested workflow, demonstrating the broadly applicable nature of our framework and methodology.

A key aspect of the proposed concept is its implementation, which delivers a tool that is both user-friendly and functionally tailored to the needs of the cryo-ET community. TANGO is a publicly available, fully open-source, Python-based tool designed to be accessible to experienced users and newcomers to the field, including those with limited programming skills, thereby supporting a wide range of expertise levels. On the one hand, it is enhanced by a user-friendly GUI that enables analyses without writing any code. This includes building custom descriptors from the features provided within the feature catalog. On the other hand, its modular design allows more experienced users to extend the provided functionality by their own features, supports, or even entire descriptors. Overall, we showed that TANGO is a versatile framework for analyzing particles from cryo-ET and has the potential to become an essential tool for elucidating the complex molecular choreography within cells.

## Methods

### Mathematics of twist vectors

In mathematical terms, orientations can be described by rotation matrices $\omega \in SO(3)$[45], where the special orthogonal group $SO(3)$ consists of orthogonal matrices of determinant 1:

$$SO(3) = \left\{ \omega \in \mathbb{R}^{3 \times 3} \mid \omega^T = \omega^{-1}, \det(\omega) = 1 \right\}. \tag{1}$$

An object characterized by coordinates centered around the origin **0** in 3D Euclidean space $\mathbb{R}^3$ can thus be rotated by having a matrix $\omega \in SO(3)$ act on each of these coordinates. The object itself remains rigid; it is neither sheared nor scaled. By combining this action with a translation as described by a vector $\mathbf{p} \in \mathbb{R}^3$, transporting the object so that its coordinates are centered around $\mathbf{p}$, results in a rigid motion[46]. A graphical depiction of such a motion can be found in Supplementary Fig. 1e. In this sense, a particle can be associated with an element of the group of rigid motions $(\omega, \mathbf{p}) \in SE(3)$, which, as a manifold, is the product $SO(3) \times \mathbb{R}^3$. In matrix form, $(\omega, \mathbf{p})$ can be explicitly handled as

$$\begin{pmatrix} \omega & \mathbf{p} \\ 0 & 1 \end{pmatrix}. \tag{2}$$

The idea of an extrinsic frame of reference to a canonical particle orientation has a natural equivalent in this context. Among rigid motions, there is a distinguished point $\mathbb{1}_4 \in SE(3)$, the $4 \times 4$ identity matrix, which can be interpreted as a canonical particle, the one positioned at the origin of $\mathbb{R}^3$ equipped with the orientation $\mathbb{1}_3 \in SO(3)$. By abuse of notation, $\mathbb{1}$ is used to denote the identity matrix when the dimension is implied.

The philosophy of using rigid motions and particles synonymously implies the existence of a product of particles and an inverse of particles. Biologically speaking, this is, of course, nonsensical. However, from a computational point of view, the idea of aligning supports is rooted in such operations.

The rigid motions of $SE(3)$ act on particle lists (finite subsets of $SE(3)$) in the following way:

$$(\omega, \mathbf{p})^{-1} \cdot (\omega, \mathbf{p}) = \mathbb{1}_4 \tag{3}$$

$$(\omega_1, \mathbf{p}_1)^{-1} \cdot (\omega_2, \mathbf{p}_2) = (\omega_1^T \omega_2, \omega_1^T(\mathbf{p}_2 - \mathbf{p}_1)) \tag{4}$$

Having the inverse of a query particle $(\omega_1, \mathbf{p}_1) \in SE(3)$ act on a collection of particles as described in Eq. 4 is equivalent to shifting and rotating the collection so that the query particle aligns with $\mathbb{1}$, compare Eq. 3, keeping the relative positions and orientations unchanged.

The basic twist descriptor satisfies this condition in a natural way. It can be constructed by considering not only the algebraic structure of $SE(3)$, but also its geometry, for particle data analysis. The group of rigid motions is a smooth manifold, which implies that neighborhoods around each point $x \in SE(3)$ can be linearly approximated through the tangent space $T_x SE(3) \cong \mathbb{R}^6$[47]. Of special interest for us is the tangent space at the distinguished point: $\mathfrak{se}(3) := T_\mathbb{1} SE(3) = T_\mathbb{1} SO(3) \times \mathbb{R}^3 = \mathfrak{so}(3) \times \mathbb{R}^3$. The reasoning behind this can once more be traced back to the desire to align supports. Finally, the space $\mathfrak{se}(3)$ is considered as the space of twist vectors

$$\xi = \begin{pmatrix} 0 & -\zeta_z & \zeta_y & p_x \\ \zeta_z & 0 & -\zeta_x & p_y \\ -\zeta_y & \zeta_x & 0 & p_z \\ 0 & 0 & 0 & 0 \end{pmatrix} \in \mathfrak{se}(3). \tag{5}$$

Note that any twist vector $\xi$ splits into a tangent $\zeta$ to $SO(3)$ at $\mathbb{1}$, given by a skew-symmetric matrix, and a vector $\mathbf{p} \in \mathbb{R}^3$. The direction $\frac{1}{\|\zeta\|}\zeta \in \mathbf{S}^2$ is the corresponding axis of rotation. The coordinates

$(\zeta_x, \zeta_y, \zeta_z, p_x, p_y, p_z) \in \mathbb{R}^6$ are so-called twist coordinates[29]. They provide a natural way to characterize the geometric interplay between a pair of particles. Given $(\omega_i, \mathbf{p}_i) \in SE(3)$, $i = 1, 2$, Eq. 4 describes their relative pose, $(\omega_1^T \omega_2, \omega_1^T(\mathbf{p}_2 - \mathbf{p}_1)) =: (\omega, \mathbf{p})$. The twist vector encoding the relevant information is the tangent vector at $\mathbb{1}$ pointing in the direction of $(\omega, \mathbf{p})$. Using the matrix logarithm $\log : SO(3) \to \mathfrak{so}(3)$[48], the twist in question is then given by $(\log(\omega), \mathbf{p})$.

Further information to be gained from these tangent vectors are their norms, which can be interpreted as geodesic distances. In the case of the positional part, this yields the Euclidean distance. For the orientational part, the geodesic distance is also known as angular distance[49] and is given by

$$d_{SO(3)}(\omega_1, \omega_2) = \| \log(\omega_1^T \omega_2)^\vee \|, \tag{6}$$

where $\cdot^\vee$ denotes the 3D vector corresponding to the input skew-symmetric matrix. The overall norm of the twist vector is the product metric for $SE(3)$.

### Angular score

Given a symmetry group from the cases discussed in our framework ($C_n$-symmetry for $n > 1$, symmetry groups of any platonic solid), the underlying idea for an unambiguous way to distinguish between orientations is to consider the geometric object that is fixed by the respective symmetry group instead of the explicit symmetries themselves. Without loss of generality, each of these objects (regular $n$-gons in the case of $C_n$-symmetry, or platonic solids) can be inscribed in a unit sphere of the appropriate dimension ($S^1 \subset \mathbb{R}^2$ for $n$-gons, $S^2 \subset \mathbb{R}^3$ for platonic solids). For every class among these objects, a representative with vertices $V \subset S^N$, $N = 1, 2$, can be chosen. As we will show later, the $C_n$-case is independent of this choice. However, for platonic solids, this is not the case. Then, given a particle of the respective symmetry, a choice of representative orientation $\omega \in SO(3)$ acts on $V$ as $\omega V = \{\omega \mathbf{v} \mid \mathbf{v} \in V\}$. Special care is required in the $C_n$-case. Here, $\omega$ can be decomposed into a product of the in-plane portion and the cone-portion. The in-plane portion $\omega_{\text{in-plane}}$ can then be projected to the group of rotations of the plane, $SO(2) \cong S^1$, to act on $V \subset S^1$. Given any other rotation $\omega'$ that represents the same orientation of the symmetric particle, we have $\omega V = \omega' V$, motivating the definition of a function to distinguish between the orientations of two particles $(\omega_i, \mathbf{p}_i)$, $i = 1, 2$, of the same symmetry type $V$:

$$\delta_V(\omega_1, \omega_2) := d_H(\omega_1 V, \omega_2 V), \tag{7}$$

where $d_H$ denotes the $S^N$-Hausdorff distance. In the case presented here, Eq. 7 simplifies to

$$\delta_V(\omega_1, \omega_2) = \max_{\mathbf{v} \in V} d_{S^N}(\omega_1 \mathbf{v}, \omega_2 V) \tag{8}$$

where $d_{S^N}(\cdot, \cdot)$ denotes the great-circle distance on $S^N$, and for a subset $W \subset S^N$ and a point $\mathbf{x} \in S^N$, $d_{S^N}(\mathbf{x}, W) = \inf_{\mathbf{w} \in W} d_{S^N}(\mathbf{x}, \mathbf{w})$.

Note that $\delta_V : SO(N+1) \times SO(N+1) \to \mathbb{R}$, $N = 1, 2$, does not define a metric since positivity does not hold. Due to rotations being isometries on spheres, the $S^N$-Hausdorff-distance $d_H$ remains unchanged under their actions, thus

$$\delta_V(\omega_1, \omega_2) = \delta_V(\omega \omega_1, \omega \omega_2) \tag{9}$$

for all $\omega \in SO(N+1)$.

If $V$ were to represent $C_n$-symmetry, commutativity of $SO(2)$ implies that $\delta_V$ is independent of the choice of regular

$n$-gon inscribed in $S^1$:

$$\begin{aligned}
\delta_{\omega V}(\omega_1, \omega_2) &= \delta_V(\omega_1\omega, \omega_2\omega) \\
&= \delta_V(\omega\omega_1, \omega\omega_2) \\
&= \delta_V(\omega_1, \omega_2)
\end{aligned} \tag{10}$$

The situation for platonic solids can be remedied by supplying a fixed orientation to act on the given vertex set $V$ in order for it to fit the user's data.

To obtain a normalized angular score, consider for each symmetry type $V$ the maximum value of $\delta_V$,

$$m_V := \max_{\omega_1, \omega_2 \in \mathrm{SO}(N+1)} \delta_V(\omega_1, \omega_2). \tag{11}$$

We can thus define an angular score for particles of symmetry type $V$ as

$$\sigma_V : \mathrm{SO}(N+1) \times \mathrm{SO}(N+1) \to [0,1], \sigma_V(\omega_1, \omega_2) = 1 - \frac{\delta_V(\omega_1, \omega_2)}{m_V}. \tag{12}$$

## Implementation details

The particle class contains necessary information for its processing, such as position, orientation, and symmetry as well as corresponding functionalities.

The twist descriptor class takes either an existing twist descriptor object or a particle list along with the initial support radius. The radius for the initial support can be derived from a statistical overview of nearest neighbor distances.

The support class takes in a twist descriptor object and computes an intersection between itself and the twist positions, removing the particles outside the support. Different supports, such as spheres, cylinders, ellipsoids, cones, and tori, are implemented as children of the support class and can be used in arbitrary orientations. Additionally, thanks to the canonical orientation of the support, one can use custom binary masks of any shape (specified as a 3D array within python or loaded from an MRC or EM file).

Filters are based on similar principles as supports. They operate on descriptor objects and remove particles from the support based on required properties. For instance, one can set an allowed range for the angular distance from the query particle and filter out all particles from the support outside this range.

Features take in twist descriptor objects that have potentially already been reduced by the support or filtering, if wanted, and compute attributes belonging to each query particle. Unlike for the twist descriptor, where each query particle can have different numbers of neighboring particles within its support, the outcome of any feature has to have the same number of values for each query point. This ensures that features can always be combined to form a new custom descriptor.

The custom descriptor class takes in a twist descriptor object, a list of desired features, and optionally a different support and a list of filters to apply. If provided, the support is first used to reduce the twist descriptor object, followed by filtering, and finally the specified features are computed for each query point. Descriptors can be easily extended by new features (or reduced by the removal of some of them), and they can also be merged with other descriptors that were computed on different supports. Besides the custom descriptor, additional descriptors are offered as well, see Supplementary Section 2 for more details.

The presented modularity enables an easy addition of new features, supports, and filters directly by users. Users can simply design, e.g., a new feature by inheriting the Feature class and ensuring compliance with inputs and outputs as described above. Such a feature will automatically be added to the GUI as well, thanks to the feature catalog

class that automatically collects all classes stemming from the parent Feature class. The same holds for supports and filters.

The computational cost for twist descriptors is influenced by the density of particle positions and the choice of search radii. Even for initial twist vector computations on data with a particularly high particle density, subsequent feature applications are efficient due to simple, fast data frame operations. Thus, analyses can easily be run locally. For user convenience, the twist descriptor can be saved into a file and loaded at a later time if needed.

## $F_1$-score

The $F_1$-score is computed as follows:

$$F_1 = \frac{2\mathrm{TP}}{2\mathrm{TP} + \mathrm{FP} + \mathrm{FN}} \tag{13}$$

where TP is the number of true positives, FP refers to the number of false positives, and FN to the number of false negatives.

## Pattern recognition - Chromatin

**Synthetic particle list.** In a modular approach to generating synthetic chromatin data, maps from EMDB corresponding to some arrangements of nucleosomes were used as building blocks: helically arranged tetranucleosomes (EMD-2601), a pair of stacked nucleosomes (EMD-13365), and a trinucleosome (EMD-13363). A depiction of one particle list corresponding to each nucleosome arrangement is shown in Supplementary Fig. 6. Given these building blocks, copies of each of them were shifted and rotated at random before being assembled into an output list. To the latter, randomly generated mononucleosomes were added. In doing so, we ensured that particles did not overlap. To this end, cryoCAT's shape-based cleaning can be applied if necessary.

**Noise corruption.** For analyses regarding synthetically generated data and robustness of procedures, there is an option to add noise to a given object of that class. This can be orientational noise, positional noise, or both. In order to add orientational noise to a particle's associated rotation, a randomly generated matrix is scaled with respect to some input noise level and added to the rotation matrix associated to the particle's orientation. Singular value decomposition ensures a resulting matrix with the characteristics of a rotation matrix. In the case of positional noise, a vector is sampled from a normal distribution centered at the origin with standard deviation (STD) given by an input noise level. This vector is then added to the particle's position.

**Dinucleosome descriptor.** To identify common axes of rotation, we considered the orientational components ($\zeta$) of the twist vectors (Eq. 5). After normalization, these vectors were mapped onto the unit sphere $S^2$ and binned to detect the most frequent rotation axis $\tilde{\zeta} \in S^2$. Deviation from this axis was quantified as $\frac{\zeta \cdot \tilde{\zeta}}{\|\zeta\|}$ and incorporated as a feature of the custom descriptor.

## Pattern recognition - Ribosomes

Following Xing et al., ribosome centers were shifted to the mRNA entry and exit points (Supplementary Fig. 10), yielding entry and exit particle lists for each condition. Twist descriptors were then computed using exit particles as queries and entry particles as neighbors.

The SHOT descriptor was computed as described in Supplementary Section 2.3 (see also Supplementary Fig. 11c). Spherical supports around each query particle were subdivided into six cones, yielding a set of four recurring occupancy patterns (SHOT footprints). These were sorted by frequency, starting with the cone aligned with the intrinsic $y$-axis of the query particle.

In the untreated dataset, the most common footprint occurred in 4968 out of 38,928 query particles, and in the treated dataset in 1554 out of 38,641.

## Reporting summary

Further information on research design is available in the Nature Portfolio Reporting Summary linked to this article.

## Data availability

No primary data was produced for this study. All STA maps from the supplementary information are available on Zenodo [https://doi.org/10.5281/zenodo.17843887]. The EMDB entries that were used for the visualization in this study are EMD-51626 [https://doi.org/10.1016/j.cell.2024.12.008] (in situ human cytoplasmic ring subunit of nuclear pore complex), EMD-63079 [https://doi.org/10.1111/gtc.70019] (native HeLa nucleosome in poly-nucleosomes (class 1)), EMD-13365 [https://doi.org/10.1038/s41594-022-00768-w] (nucleosome stack of the 4×187 nucleosome array containing H1), EMD-13363 [https://doi.org/10.1038/s41594-022-00768-w] (trinucleosome of the 4×177 nucleosome array containing H1), EMD-2601 [https://doi.org/10.1126/science.1251413] (double helix twisted by tetranucleosomes), EMD-16721 [https://doi.org/10.1126/science.adh1411] (human 80S ribosome), EMD-10398 [https://doi.org/10.15252/embj.2019103365] (yeast di-ribosome stalled on poly(A) tract). Furthermore, tomograms reconstructed from tilt series with accession code EMPIAR-10277 were used for STA of HIV-1 CA-SP1 VLPs, and a tomogram reconstructed from tilt series with accession code EMPIAR-11538 was used for STA of microtubules. The source data underlying Figs. 2d, e, 3f, 4c, d, e, f, 5c, d, e, g, h, 6, Supplementary Fig. 7, and Supplementary Tables 3 and 4 are provided as a Source Data file. Source data are provided with this paper.

## Code availability

The framework TANGO is part of the cryoCAT (Contextual Analysis Tool for cryo-electron tomography) public GitHub repository[30], specifically at [https://github.com/turonova/cryoCAT/blob/main/cryocat/tango.py]. Tutorials in the form of commented Jupyter notebooks are available under [https://github.com/turonova/cryoCAT/tree/main/docs/source/tutorials/tango_tutorials/], guiding the user through two analysis cases. The required data is provided there as well.

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

## Acknowledgements

We would like to thank Jan Philipp Kreysing for providing particle lists, score maps, and tomograms on mature HIV and NPCs as well as for helpful discussions. Huaipeng Xing for providing data for the analysis and visualization of ribosomes. Eunyoung Jeong and Jan Philipp Kreysing for testing the GUI. Eunyoung Jeong and Jiasui Liu for providing data for further testing of our approaches. Bernhard Hampölz, Desislava Glushkova, Giulia Tonon, Jan Philipp Kreysing, Maarten Tuijtel, Martin Beck, Patrick Hoffmann, and Sonja Welsch for critically reading the manuscript. Finally, we would also like to thank all members of the Beck department for sharing with us their struggle with data analysis that ultimately lead to this study. The authors received funding from the Max Planck Institute of Biophysics, Frankfurt am Main, Germany.

## Author contributions

M.S. and B.T. conceived the study. M.S. and B.T. designed and implemented the method. M.S. and B.T. performed the analyses for the use cases. M.S. and B.T. wrote the manuscript. B.T. supervised the project.

## Funding

## Competing interests

The authors declare no competing interests.
