## [Transparent Peer Review file · Nature Communications]

TANGO: Analysis and Curation of Particles in Cryo-Electron Tomography

Corresponding Author: Dr Beata Turoňová

Version 0:

Reviewer comments:

Reviewer #1

(Remarks to the Author)

Review of "Twist and Scout: Analysis and Curation of Particles in Cryo-Electron Tomography Using TANGO" by Markus Schreiber and Beata Turonova

The Authors present a new tool for cryo-electron tomogram analysis. During the process of subtomogram averaging for high resolution structure determination, the position and orientation of each particle is determined. Statistical analysis of the relative positions and orientations of particles can help us understand particle organization and functional attributes. While such analyses have been performed in the past, the authors build a tool that will be more broadly applicable to particles that may be arranged with different behaviors or different geometries. As such, the TANGO tool imports particle position and orientation lists from many common cryoET software packages and uses these point clouds to perform spatial analysis with twist vectors. The manuscript goes through test cases that include nuclear pore complexes, immature HIV-1, microtubules, mature HIV-1 capsids, and synthetic chromatin data, and ribosomes. The tool can be used for cleaning an existing particle list, understanding lattice structure, finding angular patterns. I think TANGO has the potential to be a useful addition to the cryoET toolbox. However, I found both the manuscript and the software to be not easily accessible in their current state.

1) This manuscript is not very concise. The introduction has two verbatim repeated sentences:

"TANGO is not only a framework, but it also comes with an open-source Python-based implementation which is further enhanced by a graphical user interface (GUI)."

"The outcome of the feature vector computation offers multiple possibilities for further data analysis"

Beyond this obvious issue, there is redundancy everywhere and not a clear, crisp, flow to any of the ideas. This makes it very difficult for readers to understand that it can be a simple tool for analyzing their particle lists. Please rewrite much more concisely.

2) I was hoping to be able to test this tool and give feedback on the GUI and usability. However, I was met with several issues. First, Numpydoc appears to be required but is not a part of the package or listed as a dependency. Second, this package doesn't support RELION5. Given that RELION5 has substantially new tools for cryoET in comparison to RELION 4, and is being widely used, it seems very applicable to add this compatibility. I have some old RELION 4 star files but was also unable to get those to load. Please try to fix these issues before I can thoroughly review the contents of the program.

(Remarks on code availability)

Reviewer #2

(Remarks to the Author)

This manuscript by Schreiber and Turonova describes TANGO (Twist-Aware Neighborhoods for Geometric Organization), a novel framework for using particle positions and orientations from cryo-ET to analyze the spatial relationships between particles. This framework starts with the calculation of twist vectors, which encode the spatial and orientational relationships between pairs of particles within a given area (termed a support). The set of twist vectors within a support form a twist descriptor, which encode the local neighborhood information around each particle. Filters can be used to remove particles from descriptors; descriptors can then be used to calculate features which can be used for particle analysis.

The authors show several types of data analyses that can be performed using TANGO, including assigning particles to

discreet objects and cleaning false positives that are inconsistent with expected features, analyses of lattices to find symmetry elements or disordered regions, and pattern recognition to find repeating arrangements of tethered particles.

Overall, I think the work describe here is of high quality and provides generalizable tools for particle analysis that fill a major gap in the cryo-ET field. However, the current manuscript is quite technically dense, even in the descriptions of applications, and I think a number of changes are necessary to make it more accessible to potential users. My comments are as follows:

Most of the 2D diagrams and plots have line weights that are too small and difficult to read. This is essentially true of all main figures except for figure 2; the dotted lines in figure 1 are virtually invisible on the printed page and require several fold magnification on screens. The axes labels on Fig 1, right panel, are also so small that they are pixelated. The authors should either resize figures or increase line weights or font sizes to ensure legibility.

Line 433 starts a paragraph describing “features”, but the first sentence says “Features take in twist descriptor objects... and compute the features...” effectively creating a circular logic. I think a concise description of the various levels of descriptors would greatly clarify the TANGO framework, perhaps in the form of a hierarchical tree or a table.

The term “junk particles” appears several times in the manuscript. I think a some definition of the term would make things clearer, e.g. are all junk particles false positives?

For section 3.1.1, there are several points that I find unclear. The NPC SU positions are shifted “by a distance corresponding to the NPC radius”. Does that effectively mean that there are now 8 overlapping points on the central axis of the NPC or are they shifted to the inner radius, i.e. the inner rim of the actual pore? This is performed to make the points cluster more closely and enable the use of smaller spherical supports; why is this important? Isn't this just a scaling factor on the pairwise distance vectors; why would that affect clustering? Likewise, if twist vectors are pairwise between points, why does the size of the spherical support matter?

There is a description of a cylindrical support with dimensions in nm are given, but then the readers is referred to a reference on NPC structures. I think there should at least be some justification for these specific numbers provided in the text, which can be done in a sentence or two. Is 15.5nm the radius of an NPC? How that does that relate to the radial shift performed on the particle data?

Similarly, there are very specific numbers in describing support shapes throughout, like 3.8nm for immature HIV-1 or 2.5nm/7.3nm for microtubules. I think it's unreasonable to expect the reader to understand the reason for these numbers without some description or explanation.

Most of the analyses in 3.1 start with a spherical support followed by a shaped support (e.g. a cylinder). Is it not possible or preferred to start directly with the shaped support?

For 3.1.3, it is noted that ground truth is not available so the results were evaluated by visual inspection. Is the evaluation that the particles appear affiliated with a given microtubule or that they seem to represent actual tubulin dimers? If the latter, is it possible to show subtomogram averages, e.g. of all particles, identified particles, and junk particles? Also, there appear to be lines of particles missing; is this something TANGO can help fill in?

The end of 3.2.1 makes reference to PCA. This is the only time it is mentioned in the manuscript and it is unclear what data is being analyzed by PCA. It reads as though several types of data are put in and the PCA reveals the importance of central angles, but this needs to be described in more detail.

The first sentence of 3.3 states “The functional modules of the cell interact within its molecular sociology”; I don't understand this statement. What are “functional modules”? Likewise, the term “interaction motifs” on line 733 needs clarification.

The panels in Fig 4 are not referenced in order in the text (4B, 4D, 4E, 4C, and 4G, with no references to 4A or 4F. The letter D is also missing in the figure.

Figure 5E and 5I have “ratio” on the Y-axis; what values are they ratios of? Reading the text, it seems like these should represent percentages, which are not ratios. Also, the Y-axes in 5I should be the same as in 5E to show the steady false positive rate of stacks and single particles; at a glance they look to be at zero like in 5E.

Why is there a steady ~22% detection rate of additional and stacked particles (line 835)? How does such a high false positive rate effect analysis when the ground truth is not known?

Lines 859-860 describe that ribosome particle lists were shifted to the entry and exit sites; where rotations also applied to place these sites along Cartesian axes? Is this the “intrinsic y-axis” referred to on line 876?

Is SHOT an acronym? If so, of what?

Line 889 – 891 refer to analyses of common distances per footprint; can this be shown as some type of plot or histogram?

On line 898, by “aligned between” do you mean similar/same? I think it's unclear what the point be addressed is.

Lines 901 – 902 discuss pairs that fall into bins. How many pairs overall are there? Is 44 much bigger than 26? These numbers provide me with no sense of scale, so I don't understand what I'm supposed to be interpreting.

Lin 901 refers to a "bin marked in C". Are "marked bins" referring the outlined rectangles? This should be noted in the figure legend.

The paragraph starting on line 906 refers to 3 configurations of ribosomes in polysomes; can this be noted in Fig 6?

The histograms in Fig 6 are in radians while the models shown are annotated in degrees, so it's difficult to relate the two. It may be useful to use the same units for both.

What do the colors of the models in Fig 6 indicate? I'm assuming that the models depicted are just ribosome models given the orientation encoded by the angular binning and not averages like in Fig S5. This should be clarified in the figure caption.

To summarize, I think the work presented here is scientifically sound and will be quite useful to the cryo-ET community. The spatial and orientational information for each particle is a unique feature of cryo-ET data, but has typically only been leveraged by expert labs using ad hoc solutions. TANGO will not only open up such analyses to a broader range of users, the discovery-based methods also provide new capabilities to even experienced labs. My comments above mainly deal with the readability and accessibility of the manuscript; after addressing these comments I think the manuscript will be suitable for publication.

(Remarks on code availability)

Version 1:

Reviewer comments:

Reviewer #2

(Remarks to the Author)

In this revised manuscript, I find that the authors have adequately addressed the issues raised in my initial review. Overall, the text is much easier to read and the revised figures are clearer than the initial submission. I am happy to recommend this for publication, but have noted two errors that need to be fixed:

1. In Figure 5, it seems like panels C and D are swapped; i.e. the caption for C matches panel D and vice versa. This needs to be fixed and checked against the references in the main text.
2. Supplementary Figure 4 refers to left and right and panels a – c. The figure only has a and b, and 3 columns, making left and right confusing. It appears as though the figure is transposed and the panels mislabeled.

(Remarks on code availability)

Response to the reviewers

We thank the reviewers for their critical assessment of our work and their constructive feedback. In revising the manuscript, we put considerable effort into improving the overall flow and readability. Redundancies were removed, repeated sentences corrected, and technical detail relocated to the Methods section to keep the main text concise and accessible. We believe these changes make the manuscript easier to follow while retaining the necessary information.

We would like to point out that, in addition to the revised clean version of the manuscript, we also provide a version with highlighted changes. Given the extent of the revisions, this marked-up version may be somewhat less useful for reading. For clarity, all line numbers and figure references in the responses below refer to the clean version.

In the following, we address the reviewers' concerns point by point.

Reviewer 1:

1) This manuscript is not very concise. The introduction has two verbatim repeated sentences: "TANGO is not only a framework, but it also comes with an open-source Python-based implementation which is further enhanced by a graphical user interface (GUI)." "The outcome of the feature vector computation offers multiple possibilities for further data analysis" Beyond this obvious issue, there is redundancy everywhere and not a clear, crisp, flow to any of the ideas. This makes it very difficult for readers to understand that it can be a simple tool for analyzing their particle lists. Please rewrite much more concisely.

We thank the reviewer for this helpful comment. We carefully revised the manuscript to improve clarity and readability. In particular, we removed redundancies, eliminated repeated sentences, and substantially shortened the main text up to the discussion. To improve the flow, we also moved much of the technical detail into the Methods Section, leaving the main part more concise and accessible.

While we sincerely hope that our work will be received as a practical tool for the community, we would also like to emphasize that it is more than a particle list editor. It is based on point descriptors, applied here as a mathematical framework for cryo-ET that enables particle arrangements to be analyzed in their structural and biological context. This extends the possibilities for analysis beyond list editing alone.

2) I was hoping to be able to test this tool and give feedback on the GUI and usability. However, I was met with several issues. First, Numpydoc appears to be required but is not a part of the package or listed as a dependency. Second, this package doesn't support RELION5. Given that RELION5 has substantially new tools for cryoET in comparison to RELION 4, and is being widely used, it seems very applicable to add this compatibility. I have some old RELION 4 star files but was also unable to get those to load. Please try to fix these issues before I can thoroughly review the contents of the program.

We apologize for the inconvenience and thank the reviewer for pointing this out. The missing dependency and the issue with loading RELION 4 star files have now been fixed. In addition, we have added support for RELION 5 and successfully tested it on the tutorial data provided by the authors of RELION.

We would like to note, however, that the RELION 5 star file format appears to be more complex than in previous versions. For example, Warp2/M produced RELION 5 outputs that differ from the tutorial example, although both now work with our implementation. We have tried to accommodate such differences, but since our access to RELION 5 files is limited, we cannot guarantee full robustness for all possible formatting variants. That said, if users encounter issues loading a specific file, they are encouraged to contact us via the GitHub issues page and share the file format. This will allow us to resolve the problem and further improve the software.

Reviewer 2:

Overall, I think the work describe here is of high quality and provides generalizable tools for particle analysis that fill a major gap in the cryo-ET field. However, the current manuscript is quite technically dense, even in the descriptions of applications, and I think a number of changes are necessary to make it more accessible to potential users. My comments are as follows:

We thank the reviewer for their positive assessment of the quality and potential impact of our work. We also appreciate the emphasis on accessibility for users. In addition to the structural changes made throughout the manuscript, we paid particular attention to simplifying the descriptions of applications and clarifying the presentation of results, with the goal of making the framework easier to follow for a broader readership. Below, we address the reviewer’s specific comments point by point.

Most of the 2D diagrams and plots have line weights that are too small and difficult to read. This is essentially true of all main figures except for figure 2; the dotted lines in figure 1 are virtually invisible on the printed page and require several fold magnification on screens. The axes labels on Fig 1, right panel, are also so small that they are pixelated. The authors should either resize figures or increase line weights or font sizes to ensure legibility.

We thank the reviewer for this helpful observation. We have revised the figures to improve their readability by adjusting line weights and font sizes where possible. At the same time, we followed the official Nature Portfolio requirements for figure preparation, which specify relatively small font sizes, particularly for graphs.

For Figure 1, we would also like to note that the graphs are intended for illustration of the workflow rather than for quantitative interpretation. The axis labels were therefore not essential, and to avoid distraction from the main message of the figure, we have removed them in the revised version.

Line 433 starts a paragraph describing “features”, but the first sentence says “Features take in twist descriptor objects... and compute the features...” effectively creating a circular logic. I think a concise description of the various levels of descriptors would greatly clarify the TANGO framework, perhaps in the form of a hierarchical tree or a table.

We thank the reviewer for pointing this out. We hope that our extensive rewriting of the manuscript has clarified the description of descriptors and resolved the circular logic noted in the previous version. To further improve clarity, we have added a diagram that provides a compact overview of the framework (see Supplementary Figure 2).

The term “junk particles” appears several times in the manuscript. I think a some definition of the term would make things clearer, e.g. are all junk particles false positives?

In the revised manuscript, we clarified that the first occurrence of “junk particles” refers specifically to false positives. The second occurrence has been rephrased to more accurately reflect the results shown in the referenced supplementary figure.

For section 3.1.1, there are several points that I find unclear. The NPC SU positions are shifted “by a distance corresponding to the NPC radius”. Does that effectively mean that there are now 8 overlapping points on the central axis of the NPC or are they shifted to the inner radius, i.e. the inner rim of the actual pore? This is performed to make the points cluster more closely and enable the use of smaller spherical supports; why is this important? Isn’t this just a scaling factor on the pairwise distance vectors; why would that affect clustering? Likewise, if twist vectors are pairwise between points, why does the size of the spherical support matter?

In our approach, the particles were shifted along the intrinsic $-x$ axis, but the shift was slightly smaller than the expected NPC radius. This ensured that subunits belonging to the same ring were brought closer together, rather than overlapping exactly. Because the shift was applied along the intrinsic axes, it is not equivalent to a simple scaling operation: correctly oriented particles cluster more closely, while randomly oriented ones are displaced in different directions. We added Supplementary Figure 3 to illustrate this shift and its impact on clustering.

Regarding the size of the spherical support: although twist descriptors are computed pairwise, a larger support results in more pairs per query point. Since query points are analyzed based on their neighborhood properties, an excessively large support can dilute meaningful relationships with irrelevant ones. The support should therefore be chosen to capture the neighborhood relevant for the task at hand. In this case, we used subunit orientations to connect particles based on C_8 -symmetry, meaning the support should be large enough to include all subunits from the same ring, but not extend far beyond. Without the shift, the required support would be much larger and would include many additional particles, which could obscure the common orientation signal. We have clarified this motivation in the revised text, and we hope the new supplementary figure helps illustrate the rationale more clearly.

There is a description of a cylindrical support with dimensions in nm are given, but then the readers is referred to a reference on NPC structures. I think there should at least be some justification for these specific numbers provided in the text, which can be done in a sentence or two. Is 15.5nm the radius of an NPC? How that does that relate to the radial shift performed on the particle data?

We agree that the justification of the numbers used at different steps should be clearer. In the revised version, we added depictions of all examples (NPC, HIV VLP, HIV mature capsid, and microtubule) directly in the main figures, including their physical sizes and/or relevant distances. To improve the flow of the manuscript, we have also moved the detailed numerical values from the main text to Supplementary Table 1, which now provides an overview of all parameters used.

Similarly, there are very specific numbers in describing support shapes throughout, like 3.8nm for immature HIV-1 or 2.5nm/7.3nm for microtubules. I think it’s unrea-

sonable to expect the reader to understand the reason for these numbers without some description or explanation.

As mentioned in our response above, we now include depictions of all examples in the main figures together with their physical sizes and relevant distances, so that the rationale for these numbers is clear.

Most of the analyses in 3.1 start with a spherical support followed by a shaped support (e.g. a cylinder). Is it not possible or preferred to start directly with the shaped support?

From a computational standpoint, spherical supports can be generated very efficiently using a k-d tree with a distance threshold. The use of non-spherical supports becomes efficient only once the points within the support are rotationally invariant, i.e. after the canonical orientation has been established. From a data analysis perspective, spherical supports also offer flexibility, as the directionality of the data may not be known initially. While it is in theory possible to start directly with another support shape, in practice this is less efficient or less practical to implement.

For 3.1.3, it is noted that ground truth is not available so the results were evaluated by visual inspection. Is the evaluation that the particles appear affiliated with a given microtubule or that they seem to represent actual tubulin dimers? If the latter, is it possible to show subtomogram averages, e.g. of all particles, identified particles, and junk particles? Also, there appear to be lines of particles missing; is this something TANGO can help fill in?

We thank the reviewer for this suggestion. We performed STA on the three lists as proposed and included the results in Supplementary Figure 4. The manuscript was adapted and now reads:

To further validate the procedure, we also performed STA on three datasets: the combined particle set, the cleaned list, and the list of removed particles. Initial averages were generated from TM positions and refined through 10 iterations of alignment (see Supplementary Figure 4 and Supplementary Table 2. As expected, all initial averages resembled the template to some degree. For the combined and removed lists, however, the maps degraded with further alignment, while the cleaned list improved, supporting the effectiveness of the cleaning. Some resemblance to microtubules remained in both the combined and removed sets; in the former case this is expected due to the presence of true particles in the list, while in the latter case it may indicate that a small fraction of true particles were marked as false negatives.

Regarding the missing particles: in principle, they could be added by defining an ideal footprint for a subunit and then using the SHOT descriptor to compare against existing particles. This would allow identification of subunits with different footprints and highlight where particles are missing. At present, TANGO supports the comparison step, but automated addition of missing particles is not yet implemented in the GUI (though it would be feasible in the Python environment). We hope to include this functionality in the future and have noted it in the limitations section. Any addition of particles should be validated (e.g., by STA), since missing particles may reflect either low signal (e.g., due to the missing wedge) or genuine absence in the tomograms.

The end of 3.2.1 makes reference to PCA. This is the only time it is mentioned in the manuscript and it is unclear what data is being analyzed by PCA. It reads as though several types of data are put in and the PCA reveals the importance of central angles, but this needs to be described in more detail.

We revised the text to clarify that PCA was applied to the computed feature vectors in order to identify a significant subset of features for further analysis.

The first sentence of 3.3 states “The functional modules of the cell interact within its molecular sociology”; I don’t understand this statement. What are “functional modules”? Likewise, the term “interaction motifs” on line 733 needs clarification.

We rephrased the sentence and due to its introductory character moved it to the introduction where it now reads “Cells can be understood as being organized into functional modules: groups of molecules or molecular complexes that carry out specific tasks. These modules do not act in isolation; their coordinated behavior gives rise to the overall molecular sociology of the cell.”

The panels in Fig 4 are not referenced in order in the text (4B, 4D, 4E, 4C, and 4G, with no references to 4A or 4F). The letter D is also missing in the figure.

Thank you for pointing this out. The panel letters are now fixed, the section on the HIV capsid has been restructured so that the panels are referenced in order and missing references were added.

Figure 5E and 5I have “ratio” on the Y-axis; what values are they ratios of? Reading the text, it seems like these should represent percentages, which are not ratios. Also, the Y-axes in 5I should be the same as in 5E to show the steady false positive rate of stacks and single particles; at a glance they look to be at zero like in 5E.

We thank the reviewer for pointing this out. The values shown were intended to be percentages rather than ratios, and we have corrected the labeling accordingly. The Y-axes in Figures 5e and 5i have also been adjusted to ensure consistency and to clearly show the steady false positive rate of stacks and single particles.

Why is there a steady 22% detection rate of additional and stacked particles (line 835)? How does such a high false positive rate effect analysis when the ground truth is not known?

We thank the reviewer for this comment. The steady 22% detection rate reflects nucleosomes that were randomly added to the synthetic dataset. Because of this, we cannot precisely quantify how much of this fraction truly corresponds to dinucleosome-like arrangements. To illustrate this, we provide in Figure 1 a selection of randomly chosen pairs from this group of nucleosomes.

In real data, where no ground truth is available, stricter parameters (e.g., narrower conical

Figure 1: Sample from visual inspection of mononucleosomes / stacked nucleosomes labeled as pairs. Lines indicate associations as computed by the algorithm.

supports) would be advisable to reduce false positives. In addition, subtomogram averaging can be applied to the query particles of detected pairs, using masks that extend toward the neighbors, to assess whether the observed orientations reflect biologically meaningful structures. This strategy is similar to the approach we used in the ribosome analysis. We added a note on this point to the main text.

Lines 859-860 describe that ribosome particle lists were shifted to the entry and exit sites; where rotations also applied to place these sites along Cartesian axes? Is this the “intrinsic y-axis” referred to on line 876?

This approach followed the original study by Xing et al. (Science 2023). In that work, the particles were shifted to the entry and exit sites while keeping their orientations fixed across the three lists. Any additional rotation would not be meaningful, since the entry and exit points do not have an orientation per se. The shift was therefore applied along the intrinsic axes of the ribosomes. We have added Supplementary Figure 10 to illustrate this procedure, together with relevant physiological details of the ribosome model used.

Is SHOT an acronym? If so, of what?

SHOT stands for “signature of histograms of orientations”. This is mentioned at the first occurrence of that acronym, but we also added this information in the supplementary information that talks about this descriptor in more detail.

Line 889 – 891 refer to analyses of common distances per footprint; can this be shown as some type of plot or histogram?

We added histograms showing distance distributions per footprint in Supplementary Figure 7, including an explanation on the interpretation of the footprints in the caption. Furthermore, some details on SHOT parameters were added to the Supplementary Figure 11 explaining extra descriptors.

On line 898, by “aligned between” do you mean similar/same? I think it’s unclear what the point be addressed is.

Yes, the footprints were the same, and we have rephrased the text to make this explicit (now on lines 653-656):

In all but one footprint, the most common axis of rotation was the same across conditions. The exception was the third footprint, where untreated and treated samples showed distinct preferred axes (Figure 6 c, f).

Lines 901 – 902 discuss pairs that fall into bins. How many pairs overall are there? Is 44 much bigger than 26? These numbers provide me with no sense of scale, so I don’t understand what I’m supposed to be interpreting.

To provide a clearer sense of scale, the total number of pairs for each SHOT footprint has been added to the caption and to the histograms in Figure 6. In our efforts to improve readability, these numbers are no longer included in the main text.

Line 901 refers to a “bin marked in C”. Are “marked bins” referring the outlined rectangles? This should be noted in the figure legend.

Yes, the markings were supposed to be the black outlines. An explanation was added to the caption of Figure 6.

The paragraph starting on line 906 refers to 3 configurations of ribosomes in polysomes; can this be noted in Fig 6?

Three representative pairs, one for each configuration, were marked as such and referenced in the caption of Figure 6.

The histograms in Fig 6 are in radians while the models shown are annotated in degrees, so it’s difficult to relate the two. It may be useful to use the same units for

both.

The data shown in the histograms corresponds to polar coordinates of axes of common rotations and are thus depicted in radians (following convention on polar histograms). The additional sampling refers to angular distances around these axes, which are reported in degrees for easier interpretation and to match the convention used throughout the paper. In this form, angular distance and rotation axis need not be directly linked; we have adjusted the legend of Figure 6 to make this distinction clearer.

What do the colors of the models in Fig 6 indicate? I'm assuming that the models depicted are just ribosome models given the orientation encoded by the angular binning and not averages like in Fig S5. This should be clarified in the figure caption.

In Figure 6, the colors represent different bins of angular scores, with distinct hues used to distinguish between SHOT footprints. The depicted pairs are indeed based on a ribosome model (EMD-16721) rather than averages of the pairs. We have clarified this in the figure caption.

To summarize, I think the work presented here is scientifically sound and will be quite useful to the cryo-ET community. The spatial and orientational information for each particle is a unique feature of cryo-ET data, but has typically only been leveraged by expert labs using ad hoc solutions. TANGO will not only open up such analyses to a broader range of users, the discovery-based methods also provide new capabilities to even experienced labs. My comments above mainly deal with the readability and accessibility of the manuscript; after addressing these comments I think the manuscript will be suitable for publication.

We thank the reviewer for the positive assessment and for recognizing the potential usefulness of TANGO for the cryo-ET community. We also appreciate the constructive suggestions regarding readability and accessibility, which we have addressed in the revised manuscript.

Reviewer #2 (Remarks to the Author):

In this revised manuscript, I find that the authors have adequately addressed the issues raised in my initial review. Overall, the text is much easier to read and the revised figures are clearer than the initial submission. I am happy to recommend this for publication, but have noted two errors that need to be fixed:

1. In Figure 5, it seems like panels C and D are swapped; i.e. the caption for C matches panel D and vice versa. This needs to be fixed and checked against the references in the main text.

2. Supplementary Figure 4 refers to left and right and panels a – c. The figure only has a and b, and 3 columns, making left and right confusing. It appears as though the figure is transposed and the panels mislabeled.

We thank the reviewer for their careful re-evaluation of our revised manuscript and for the positive recommendation for publication. We greatly appreciate the reviewer's constructive feedback throughout the process, which has significantly improved the clarity and quality of the work.

We are also grateful to the reviewer for spotting the remaining issues in the figure legends. We have corrected the swapped panels in Figure 5 and ensured that the references in the main text now match. In addition, Supplementary Figure 4 has been revised to remove the inconsistencies in panel labeling and orientation, and the description has been updated accordingly.